# SHARP ASYMPTOTIC THEORY FOR Q-LEARNING WITH LD2Z LEARNING RATE AND ITS GENERALIZATION

**Soham Bonnerjee**
Department of Statistics
University of Chicago
sohambonnerjee@uchicago.edu

**Zhipeng Lou**
Department of Mathematics
University of California, San Diego
zlou@ucsd.edu

**Wei Biao Wu**
Department of Statistics
University of Chicago
wbwu@uchicago.edu

## ABSTRACT

Despite the sustained popularity of Q-learning as a practical tool for policy determination, a majority of relevant theoretical literature deals with either constant ($\eta_t \equiv \eta$) or polynomially decaying ($\eta_t = \eta t^{-\alpha}$) learning schedules. However, it is well known that these choices suffer from either persistent bias or prohibitively slow convergence. In contrast, the recently proposed linear decay to zero (LD2Z: $\eta_{t,n} = \eta(1 - t/n)$) schedule has shown appreciable empirical performance, but its theoretical and statistical properties remain largely unexplored, especially in the Q-learning setting. We address this gap in the literature by first considering a general class of power-law decay to zero (PD2Z-$\nu$: $\eta_{t,n} = \eta(1 - t/n)^{\nu}$). Proceeding step-by-step, we present a sharp non-asymptotic error bound for Q-learning with PD2Z-$\nu$ schedule, which then is used to derive a central limit theory for a new *tail* Polyak-Ruppert averaging estimator. Finally, we also provide a novel time-uniform Gaussian approximation (also known as *strong invariance principle*) for the partial sum process of Q-learning iterates, which facilitates bootstrap-based inference. All our theoretical results are complemented by extensive numerical experiments. Beyond being new theoretical and statistical contributions to the Q-learning literature, our results definitively establish that LD2Z and in general PD2Z-$\nu$ achieve a best-of-both-worlds property: they inherit the rapid decay from initialization (characteristic of constant step-sizes) while retaining the asymptotic convergence guarantees (characteristic of polynomially decaying schedules). This dual advantage explains the empirical success of LD2Z while providing practical guidelines for inference through our results.

## 1 INTRODUCTION

With the advent of generative AI models and its continuing ascent towards ubiquity, the use of reinforcement learning (RL) to train multiple agents to undertake complex sequential decisions seamlessly, has occupied a central role in modern learning theory. In that regard, Q-learning (Watkins et al., 1989; Watkins & Dayan, 1992; Sutton & Barto, 2018; Chi et al., 2025), represents a classical, yet practically relevant model-free approach to estimate the optimal policy of a Markov decision process (MDP). Research on the statistical properties of the Q-learning algorithm has been extensive; in particular, treatment of asymptotic and non-asymptotic error bounds have ranged from techniques particular to synchronous Q-learning (Jaakkola et al., 1993; Tsitsiklis, 1994; Szepesvári, 1997; Shi et al., 2022), to the more modern lens of stochastic approximation (SA) algorithms (Chen et al., 2020b; Qu & Wierman, 2020; Chen et al., 2021). Specifically, these latter works cast the Q-learning algorithm as an SA targeting the Bellman equation, and thereby, more general tools can be employed to derive finer theoretical results on these algorithms. This direction also has been, arguably, adequately explored with central limit theory, and functional central-limit-theorems, appearing in (Xie & Zhang, 2022; Li et al., 2023b;a; Panda et al., 2024). A special case of Q-learning with a singleton action

space, is the Temporal-difference (TD) learning, for which Berry-Esseen theorems and subsequent Gaussian approximations and bootstrap strategies have been discussed (Wu et al., 2024b; 2025; Samsonov et al., 2025).

A very important, but often ignored aspect in these theoretical studies is the choice of step-sizes or learning rates. Indeed, it has become widely common in statistical inference literature to analyze either the constant learning rates or the polynomially decaying learning rate. Such choices are not without their own advantages; the constant learning rate enjoys experimental evidence of a much faster convergence, however a proof similar to Li et al. (2024b) shows that the Q-learning with constant learning rate will converge to a stationary distribution around the optimal $\mathbf{Q}^\star$; in other words, the asymptotic bias is non-negligible, and requires further jackknifing to ensure convergence. On the other hand, the polynomially decaying learning rate is theoretically attractive; the aforementioned results establishing Gaussian approximations and other inferential results extensively use a polynomially decaying learning rate. This choice has been guided by theory of stochastic gradient descent at least since (Ruppert, 1988; Polyak & Juditsky, 1992), however its theoretical optimality often masks its excruciatingly slow convergence, as also observed by (Zhang & Xie, 2024). These criticisms have been echoed by the broad stochastic optimization community, leading to a recent proposal of linearly decaying to zero (LD2Z) learning rate $\eta_{t,n} = \eta(1 - t/n)$ (Devlin et al., 2019; Touvron et al., 2023). Despite a requirement of pre-specified number of schedules, this step-size choice achieves a balance between the rapid initial dissipation of initialization effects provided by a constant learning rate and the asymptotic convergence guarantees of a polynomially decaying learning rate. In this article, we establish a number of sharp asymptotic results for the Q-learning algorithm with this particular learning rate schedule. To the best of our knowledge, our results are the first-of-its-kind theory using this step-size for Q-learning; the theoretical results and subsequent numerical exercise definitively showcases the effectiveness and superiority of this learning rate over the ones usually employed in theoretical analyses.

## 1.1 MAIN CONTRIBUTIONS

The paper develops a comprehensive theoretical framework for Q-learning with power-law decay to zero (PD2Z-$\nu$) learning schedules. Our results advance the theoretical understanding of Q-learning and offer new insights into its statistical properties and practical performance. The main contributions are summarized below:

- **Non-asymptotic concentration inequality.** Under standard regularity conditions, we derive explicit non-asymptotic bounds on the $p$-th moments of the Q-learning iterates for any fixed $p \geq 2$. In particular, our $\mathcal{L}_2$ bounds can be summarized as follows.

  **Theorem 1.1** (Theorem 3.1, Informal). *If $\mathbf{Q}_n$ denotes the final Q-learning iterate with the PD2Z-$\nu$ step-size, then it follows that*

  $$\|\mathbf{Q}_n - \mathbf{Q}^\star\|_2 \lesssim \exp(-cn)|\mathbf{Q}_0 - \mathbf{Q}^\star| + n^{-\frac{\nu}{2(\nu+1)}},$$

  *where $\mathbf{Q}^\star$ is the long term reward corresponding to the optimal policy $\pi^\star$.*

  These bounds serve as fundamental tools underpinning the empirical success of Q-learning with PD2Z-$\nu$ schedules compared to their polynomially decaying counterparts (Section 3.1). In particular, the exponential decay from the initialization is empirically observed in Figure 1, further validating our theory.

- **Distribution theory.** We propose a novel averaging scheme that aggregates a batch of the most recent Q-learning iterates, referred to as the *tail Polyak-Ruppert averaging estimator*, and establish its asymptotic normality (Section 3.2). This is, to the best of our knowledge, a novel contribution in stochastic approximation literature. For the PD2Z-$\nu$ learning schedules, our simulation (in §5.4) also establishes the superiority of tail PR averaged estimator over the usual PR averaged ones.

- **Strong invariance principle.** We establish strong invariance principles with covariance matching for the partial sum processes of Q-learning with both PD2Z-$\nu$ and polynomially decaying learning schedules. This is accomplished via a novel construction of the coupling Gaussian process, enabling a more refined probabilistic analysis of the stochastic dynamics (Section 4).

## 1.2 RELATED LITERATURE

Linearly decaying-to-zero (`LD2Z`) learning-rate schedules have recently gained substantial traction in applications characterized by highly non-smooth or complex optimization landscapes, including state-space models (Touvron et al., 2023), large language models (Devlin et al., 2019; Liu et al., 2019; Bergsma et al., 2025), and vision transformers (Wu et al., 2024a). A number of studies further advocate for the so-called "knee schedule" (Howard & Ruder, 2018; Hoffmann et al., 2022; Iyer et al., 2023; Defazio et al., 2023; Hägele et al., 2024; Bergsma et al., 2025), which employs an initial large learning rate (a "warm start") followed by a `LD2Z` phase. Despite their empirical popularity, the asymptotic properties of `LD2Z` schedules remain poorly understood—even in relatively simple convex problems. To the best of our knowledge, Goldreich et al. (2025) provides the first theoretical analysis of `LD2Z` schedules in strongly convex stochastic gradient descent; but their results are not directly applicable to Q-learning, and they only establish an $\mathcal{L}_2$ control of the terminal iterates $\mathbf{Q}_{n,n}$. This gap in theory presents a significant obstacle to principled statistical inference and uncertainty quantification, motivating the need for a more systematic analysis.

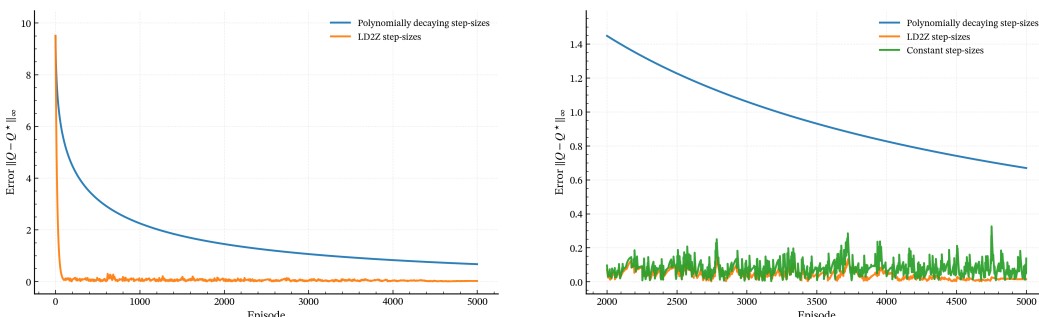

Figure 1: Comparison between polynomially decaying ($\eta_t = 0.05t^{-0.65}$), `LD2Z`($\eta_t = 0.05(1-t/n)$) and Constant ($\eta_t = 0.05$) step-sizes

## 1.3 NOTATION

In this paper, we denote the set $\{1, \ldots, n\}$ by $[n]$. The $d$-dimensional Euclidean space is $\mathbb{R}^d$, with $\mathbb{R}^d_{>0}$ the positive orthant. For a vector $a \in \mathbb{R}^d$, $|a|$ denotes its Euclidean norm. The set of $m \times n$ real matrices is denoted by $\mathbb{R}^{m \times n}$, and correspondingly, for $M \in \mathbb{R}^{m \times n}$, $|M|_F$ denotes its Frobenius norm. For a random vector $X \in \mathbb{R}^d$, we denote $\|X\| := \sqrt{\mathbb{E}[|X|^2]}$. We also denote in-probability convergence, and stochastic boundedness by $o_\mathbb{P}$ and $O_\mathbb{P}$ respectively. The weak convergence is denoted by $\xrightarrow{w}$. We write $a_n \lesssim b_n$ if $a_n \leq C b_n$ for some constant $C > 0$, and $a_n \asymp b_n$ if $C_1 b_n \leq a_n \leq C_2 b_n$ for some constants $C_1, C_2 > 0$.

## 2 PRELIMINARIES OF Q-LEARNING

Subsequently, we consider a discounted, infinite horizon Markov Decision Process (MDP) $\mathcal{M} = (\mathcal{S}, \mathcal{A}, \gamma, \mathbb{P}, R)$. Here $\mathcal{S} = \{1, \ldots, S\}$ is the *finite* state space, $\mathcal{A}$ is the finite action space, and $\gamma \in (0, 1)$ is the discount factor. For simplicity, we define $D = |\mathcal{S} \times \mathcal{A}|$. We use $\mathcal{P} : \mathcal{S} \times \mathcal{A} \to \Delta(\mathcal{S})$ to represent the probability transition kernel with $\mathcal{P}(s'|s, a)$ the probability of transiting to $s'$ from a given state-action pair $(s, a) \in \mathcal{S} \times \mathcal{A}$. Let $R : \mathcal{S} \times \mathcal{A} \to [0, \infty)$ stand for the random reward, i.e., $R(s, a)$ is the immediate reward collected in state $s \in \mathcal{S}$ when action $a \in \mathcal{A}$ is taken. We represent the distribution $\mathbb{P}(s'|s, a)$ using quantile transformation: there exists a measurable function $N(s, a, U)$, where $U \sim \text{Uniform}(0, 1)$, such that

$$\mathbb{P}(N(s, a, U) = s') = \mathbb{P}(s'|s, a) \text{ for all } s, s' \in \mathcal{S} \text{ and } a \in \mathcal{A}.$$

Similarly, we can write the reward function as $R(s, a, \mathcal{U})$, where $\mathcal{U} \sim \text{Uniform}(0, 1)$. Let $\pi$ be a policy, meaning that for each $s \in \mathcal{S}, \pi(\cdot|s)$ is a probability distribution over actions $a \in \mathcal{A}$. Define

the expected long-term reward

$$\mathbf{Q}^\pi(s,a) = \mathbb{E}^\pi \left\{ \sum_{i=0}^\infty \gamma^i R\left(s_t, a_t, \mathcal{U}_t\right) \mid s_0 = s, a_0 = a \right\}.$$

Let $\mathbf{Q}^* = (\mathbf{Q}^*_{sa})_{(s,a) \in \mathcal{S} \times \mathcal{A}}$ where $\mathbf{Q}^*_{sa} = \max_\pi \mathbf{Q}^\pi(s,a)$ is the maximizer.

To estimate $\mathbf{Q}^*$, the $Q$-function vector $\mathbf{Q}_t \in \mathbb{R}^D$ is updated by (e.g., Watkins & Dayan (1992))

$$\mathbf{Q}_{t,n} = (1 - \eta_{t,n})\mathbf{Q}_{t-1,n} + \eta_{t,n}\widehat{B}_t\mathbf{Q}_{t-1,n}, \quad \mathbf{Q}_{0,n} = \mathbf{Q}_0, \tag{2.1}$$

where $\widehat{B}_t$ is the empirical Bellman operator given by

$$(\widehat{B}_t\mathbf{Q})(s,a) = R(s,a,V_{t,n}) + \gamma \max_{a' \in \mathcal{A}} \mathbf{Q}(N(s,a,U_t),a'), \quad \mathbf{Q} \in \mathbb{R}^D. \tag{2.2}$$

Here $U_t, \mathcal{U}_t, t \in \mathbb{Z}$, are i.i.d. $\mathrm{Uniform}(0,1)$ random variables. With a slight abuse of notations, define the matrix $\mathcal{P} \in \mathbb{R}^{D \times |\mathcal{S}|}$ with rows $\mathcal{P}_{(s,a),\cdot} = (\mathcal{P}(s'|s,a))_{s' \in \mathcal{S}}^\top$. If $\Pi^\pi \in \mathbb{R}^{S \times D}$ is a projection matrix associated with a given policy $\pi$:

$$\mathbf{\Pi}^\pi = \mathrm{diag}\left\{\pi(\cdot|1)^\top, \cdots, \pi(\cdot|S)^\top\right\},$$

then we define the Markov transition kernel $H^\pi = \mathcal{P}\Pi^\pi \in \mathbb{R}^{D \times D}$.

# 3 Q-LEARNING DYNAMICS WITH LD2Z SCHEDULE AND BEYOND

Before introducing our key results on Q-learning with the LD2Z schedule and its generalization, it is crucial to state the regularity conditions that guarantee the validity of the theoretical excursion. In particular, we require the following assumptions.

**Assumption 3.1.** *It holds that* $\mathbb{E}|R(s,a)|^p < \infty$ *for all* $(s,a) \in \mathcal{S} \times \mathcal{A}$, *for some* $p \geq 2$.

**Assumption 3.2.** *There exist* $\pi^* \in \Pi^*$ *and a positive constant* $L < \infty$ *such that for any function estimator* $\mathbf{Q} \in \mathbb{R}^D$, *we have*

$$|(H^{\pi_Q} - H^{\pi^*})(\mathbf{Q} - \mathbf{Q}^*)|_\infty \leq L|\mathbf{Q} - \mathbf{Q}^*|^2_\infty,$$

*where* $\pi_Q(s) := \arg\max_{a \in \mathcal{A}} Q(s,a)$ *is the greedy policy w.r.t.* $Q$.

Assumption 3.1 establishes a uniform control over the $p$-th moment of the reward function. In contrast, often the statistical literature on this topic imposes a severely restrictive condition of a bounded reward, usually constrained in the interval $[0,1]$ or $[-1,1]$ (Li et al., 2021; Shi et al., 2022; Panda et al., 2024; Li et al., 2024a; Zhang & Xie, 2024; Chen, 2025). We also remark that Assumption 3.1 is objectively weaker than the corresponding bounded fourth moment assumption in Li et al. (2023b). On the other hand, conditions of the type of Assumption 3.2 were first introduced in Puterman & Brumelle (1979), and have since been employed in Q-learning literature (Li et al., 2023b; Xia et al., 2024) as a means to establish a *local attraction basin* around the optimal policy $\pi^\star$. Interestingly, this can also be derived from a mild margin condition, as is described in Appendix §9. The corresponding versions of Assumptions 3.1-3.2 is pervasive in non-asymptotic analysis of SA algorithms (Ruppert, 1988; Polyak & Juditsky, 1992; Borkar, 2023; Bottou et al., 2018; Chen et al., 2020a; Zhu et al., 2023; Wei et al., 2023).

## 3.1 NON-ASYMPTOTIC ERROR BOUND

Before establishing inferential results involving LD2Z schedules, it is crucial to ascertain their non-asymptotic convergence properties. On the other hand, it is conceivable to broaden our view to the class of learning schedules $\eta_{t,n} = \eta(1 - t/n)^\nu, \nu > 0$, of which LD2Z is but a special case with $\nu = 1$. This perspective raises another pertinent question; due to the lack of previous theoretical justifications, it is somewhat unclear as to why the linear decay-to-zero is less effective, in any sense, compared to some iteration-dependent choice of $\nu$. We address both the questions through our first result. For brevity, we subsequently refer to the schedule $\eta_{t,n} = \eta(1 - t/n)^\nu$ as the Power-law decay to zero (abbreviated as PD2Z-$\nu$).

Let the Bellman noise be given by

$$Z_t(s,a) = \widehat{B}_t(\mathbf{Q}^*)(s,a) - B(\mathbf{Q}^*)(s,a), \tag{3.1}$$

which, via (2.2) immediately implies that $Z_t$ are i.i.d. $D$-dimensional random vectors. Our first theorem is presented below.

**Theorem 3.1.** *Consider the Q-learning iterates in (2.1). Suppose for some $p \geq 2$, the Bellman noise satisfies $\Theta_p := \mathbb{E}[|Z_t|^p] < \infty$. Then, with the PD2Z-$\nu$ learning schedule with $\eta > 0$, $\nu \geq 1/p$ satisfying*

$$\eta < \frac{2(1-\gamma)}{(1-\gamma)^2 + 2(p-1)\gamma^2},$$

*it holds that*

$$\|\mathbf{Q}_{t,n} - \mathbf{Q}^\star\|_p \leq \exp\left(-c_3 \eta t (1-n^{-1})^\nu\right)|\mathbf{Q}_0 - \mathbf{Q}^\star|$$
$$+ 2\sqrt{p-1}\Theta_p^{1/p} \begin{cases} \sqrt{C_1(c_3,\nu,2)}\sqrt{\eta_{t,n}}, & t \leq n - \frac{2}{(c_3\eta)^{\frac{1}{\nu+1}}} n^{\frac{\nu}{\nu+1}}, \\ \sqrt{C_2(c_3,\nu,2)}n^{-\frac{\nu}{2(\nu+1)}}, & t > n - \frac{2}{(c_3\eta)^{\frac{1}{\nu+1}}} n^{\frac{\nu}{\nu+1}}, \end{cases} \tag{3.2}$$

*where $c_3 = \frac{\eta c_1 - \eta^2 c_2}{2\eta}$ with $c_1 = 2(1-\gamma), c_2 = (1-\gamma)^2 + 2(p-1)\gamma^2$, and $C_1(c,\nu,p)$, $C_2(c,\nu,p)$ are positive constants given by*

$$C_1(c,\nu,p) := \frac{2^{\nu(p+1)}(1 + 2^{-p}\Gamma(\nu p + 1))}{c}, \text{ and,}$$
$$C_2(c,\nu,p) := \eta^p 4^{\nu p} \exp\left(\frac{2^{\nu+1}}{\nu+1}\right)(\nu+1)^{(p-1)\frac{\nu}{\nu+1}}(c\eta)^{-\frac{\nu p+1}{\nu+1}}\Gamma\left(\frac{\nu p + 1}{\nu+1}\right).$$

Theorem 3.1 is proved in Appendix §7.

*Remark* 3.1 (A sample complexity version of Theorem 3.1). Let $N(\epsilon, \gamma, \nu)$ denotes the minimal number of samples required to ensure $\|Q_{n,n} - Q^*\|_q \leq \epsilon$. In the worst case, $\Theta_p^{1/p} \lesssim \frac{1}{1-\gamma}$. Therefore, from Theorem 3.1 , we obtain the following *iteration complexity*:

$$N(\epsilon, \gamma, \epsilon) = O\left(\frac{1}{(1-\gamma)^2}\log\left(\frac{|Q_0 - Q^*|}{\epsilon}\right) + \frac{1}{(1-\gamma)^{4+2/\nu}\epsilon^{2(\nu+1)/\nu}}\right).$$

We note that for large $\nu$, the rate approximately matches that derived by Li et al. (2024a). The gap for a finite value of $\nu$ can also be explained by the much weaker assumption that we work with. For example, we do not assume the rewards to be bounded, and therefore, are only constrained to work with finite $p$-th moments of the Bellman noise. In contrast, Li et al. (2024a) assumes the rewards $\in [0, 1]$, which makes the Bellman noise sequences bounded and allows them to use finer tools from subGaussian theory, such as Freedman's inequality (in contrast to the Burkholder's inequality which is sharp in absence of boundedness). It is conceivable that in presence of stricter assumption, the worst-case sample complexity can be further improved, but that is non-trivial.

The non-asymptotic bound in (3.2) is convenient since it covers a general class of learning schedules with an explicitly quantified bound. Crucial is also the two distinct regimes with two different rates. We pause for a moment to parse the bound carefully. In the *transient regime* with $t \leq n - C_{\eta,\nu} n^{\frac{\nu}{\nu+1}}$, the $\mathcal{L}_2$ error decays with $\eta_{t,n}$. In particular, for any choice of $\nu > 0$, $\eta_{t,n} \asymp 1$ as long as $t \leq nc$ for any fixed constant $c \in (0, 1)$. Therefore, in the early regime, the class of PD2Z-$\nu$ learning schedules behave like a constant learning rate while decaying polynomially. The corresponding $\mathcal{L}_2$ error displays a diminishing bias, but this constant learning rate is a crucial key to its much faster convergence, pushing it towards its *convergence regime* where $t > n - C_{\eta,\nu} n^{\frac{\nu}{\nu+1}}$. In this regime the Q-learning chain has converged with an error-rate $n^{-\frac{\nu}{2(\nu+1)}}$, enabling an early stopping at any steps in $[n - C_{\eta,\nu} n^{\frac{\nu}{\nu+1}}, n]$.

The afore-mentioned fast decay, followed by a stabilization in the latter phase, is exemplified empirically in Figure 1. For a more detailed insight into this early phase decay, it is instrumental to specify one immediate corollary to Theorem 3.1.

**Corollary 3.2.** *Under the assumptions of Theorem 3.1, it follows that for all $t \in [n]$,*

$$\|\mathbf{Q}_{t,n} - \mathbf{Q}^\star\|_p \leq \exp\big(-c_3\eta(1 - n^{-1})^\nu t\big)|\mathbf{Q}_0 - \mathbf{Q}^\star| + O_{c_3,\nu}(\sqrt{\eta_{t,n}} \vee n^{-\frac{\nu}{2(\nu+1)}}),$$

*where $O_{c_3,\nu}$ hides constants pertaining to $c_3$ and $\nu$. We note that at $t = n$, the right hand side is minimized at $\nu \asymp \log_2 \log n$.*

Corollary 3.2 has some interesting connotations, which we will discuss in successive remarks. To initiate our first discussion, it is illuminating to recall the following well-known result for the often-used polynomially decaying learning schedules.

**Theorem 3.3** (Chen et al. (2020b), Corollary 4.1.2; Li et al. (2023b), Theorem E.1). *Consider the Q-learning iterates in (2.1) with the polynomially decaying step-size $\eta_t \asymp t^{-\alpha}$, $\alpha \in (1/2, 1)$. Then, it follows that for all $t \in [n]$,*

$$\|\mathbf{Q}_t - \mathbf{Q}^\star\|_p \lesssim \exp(-ct^{1-\alpha})|\mathbf{Q}_0 - \mathbf{Q}^\star| + O(t^{-\alpha/2}).$$

In light of Theorem 3.3, Corollary 3.2 sheds more light on the faster decay of the `LD2Z` and in general `PD2Z`-$\nu$ schedules in the transient phase.

*Remark* 3.2. Assume $\nu > 0$ is fixed. Note that, in particular, when $t = n$, i.e. at the final iterate, Q-learning with `PD2Z`-$\nu$ schedule instructs that

$$\|\mathbf{Q}_{n,n} - \mathbf{Q}^\star\|_p \lesssim \exp(-4^{-1}n)|\mathbf{Q}_0 - \mathbf{Q}^\star| + n^{-1/4}.$$

The dominating decay rate in the *convergence phase* (the second term in the rates on the right) is similar in both `PD2Z`-$\nu$ and polynomial decay schedules ($n^{-\frac{\nu}{2(\nu+1)}}$ versus $n^{-\alpha/2}$); however, the effect of initial point is much less pronounced in the former, with an exponential rate $\exp(-ct)$ of *forgetting* the initialization for all $t \in [n]$. This explains the fast initial convergence of this linearly decaying rate to a neighborhood of $\mathbf{Q}^\star$, as also seen in Figure 1. In contrast, the polynomial step-size only achieves a *forgetfulness* of $\exp(-ct^{1-\alpha})$. This explains the competitive advantage of linearly decaying rate over its polynomial counterpart- an advantage that has also been recently studied in the empirical literature (Defazio et al., 2023; Bergsma et al., 2025). To the best of our knowledge, this is the first such theoretical exposition highlighting the benefits of linear decay rate `LD2Z` and its generalization in the context of Q-learning, while building on the previous works of Goldreich et al. (2025) in the more general context of Stochastic Approximation algorithms.

Next, we explore another interesting assertion from Corollary 3.2 regarding the optimal choice of $\nu$ in the class of `PD2Z`-$\nu$ learning schedules.

*Remark* 3.3. The optimal $\nu$ balances the fact that $C_2(c_3, \nu, 2)$ increases with $\nu$, while $n^{-\frac{\nu}{2(\nu+1)}}$ decreases with $\nu$ for large $n \in \mathbb{N}$. This trade-off yields the threshold $\nu \asymp \log_2 \log n$, which grows extremely slowly with $n$, justifying fixed, iteration-independent choices of $\nu$ in practice. This aligns with the empirical success of $\nu = 1$, motivating deeper statistical study under the assumption of constant $\nu$. In particular, to round off our discussion on choices of $\nu$, we state a clean result on Q-learning dynamics with `LD2Z` schedule.

**Corollary 3.4.** *Under the assumptions of Theorem 3.1, for the `LD2Z` learning schedule it follows that for $t \in [n]$,*

$$\|\mathbf{Q}_{t,n} - \mathbf{Q}^\star\|_p \leq \exp\big(-c_3\eta 2^{-1}t\big)|\mathbf{Q}_0 - \mathbf{Q}^\star| + \begin{cases} O(\sqrt{\eta_{t,n}}), & t \leq n - \frac{2}{\sqrt{c\eta}}\sqrt{n} \\ O(n^{-1/4}), & t > n - \frac{2}{\sqrt{c\eta}}\sqrt{n}, \end{cases} \tag{3.3}$$

*where $O(\cdot)$ hides constants depending on $\gamma$ and $\eta$.*

Subsequently, we assume that $\nu$ is fixed, and move towards sharper asymptotic result beyond $\mathcal{L}_2$ control.

## 3.2 TAIL POLYAK-RUPPERT AVERAGES AND CENTRAL LIMIT THEORY

As a means of variance reduction and faster convergence, Polyak-Ruppert averaging (Ruppert, 1988; Polyak & Juditsky, 1992) has a relatively long history of application in policy evaluation (Bhandari et al., 2018; Khamaru et al., 2021), Q-learning (Li et al., 2023a;b; 2024a) and Temporal Difference

(TD) learning (Mou et al., 2020; Samsonov et al., 2024; 2025). However, our $\mathcal{L}_2$ error-bounds reveal a crucial insight into whether usual Polyak-Ruppert averaging would ensure asymptotic normality with these LD2Z and PD2Z-$\nu$ schedules. Consider $\nu = 1$. Write

$$n^{-1} \sum_{t=1}^{n} \mathbf{Q}_{t,n} = \frac{1}{2} \frac{\sum_{t=1}^{n/2} \mathbf{Q}_{t,n}}{n/2} + \frac{1}{2} \frac{\sum_{t=n/2}^{n-\sqrt{n}} \mathbf{Q}_{t,n}}{n/2} + \frac{1}{2} \frac{\sum_{t=n-\sqrt{n}}^{n} \mathbf{Q}_{t,n}}{n/2} := A_n + B_n + C_n. \quad (3.4)$$

Observe that as long as $t \leq n/2$, it holds $\eta_{t,n} \geq \frac{\eta}{2^\nu}$. Therefore, based on the intuition from stochastic approximation literature with constant step-size, we do not expect $A_n$ to even converge to $\mathbf{Q}^\star$, let alone achieve asymptotic Gaussianity. It is not yet clear if $C_n$ may achieve Gaussianity individually; at the very least, its $\mathcal{L}_p$ convergence to $\mathbf{Q}^\star$ is guaranteed through an argument similar to Theorem 3.1. Therefore, unless one shows that the asymptotic distribution of $B_n$ exactly cancels that of $A_n$, it is conceivable that the error of $n^{-1} \sum_{t=1}^{n} \mathbf{Q}_{t,n}$ is in effect, much larger compared to $\mathbf{Q}^\star$. This theoretical insight can also be empirically validated (Figure 4). Therefore, it is arguably more prudent to investigate the inferential properties of the term $C_n$, which we refer to as *Tail Polyak-Ruppert Averages*.

**Theorem 3.5.** *For any constant $c > 0$ and $\nu \geq 1/p$ with $p \geq 2$ is same as in Assumption 3.1, let*

$$\bar{\mathbf{Q}}_n = \frac{1}{\lfloor cn^{\frac{\nu}{\nu+1}} \rfloor} \sum_{t=n-\lfloor cn^{\frac{\nu}{\nu+1}} \rfloor+1}^{n} \mathbf{Q}_{t,n}.$$

*Grant Assumptions 3.1 and 3.2 for the MDP. Further assume that $\mathbf{Q}_0, \mathbf{Q}^\star \in K$ where $K$ is a compact set. Then with the PD2Z-$\nu$ learning rate for (2.1) with,*

$$0 < \eta < \frac{2(1-\gamma)}{(1-\gamma)^2 + 2(p-1)\gamma^2},$$

*there exists a positive definite matrix $\Sigma \succeq 0$ independent of $n$, such that*

$$n^{\frac{\nu}{2(\nu+1)}} (\bar{\mathbf{Q}}_n - \mathbf{Q}^\star) \xrightarrow{w} N(0, \Sigma). \quad (3.5)$$

Theorem 3.5 is proved in Appendix §7. We remark that an exact expression for $\Sigma$ is highly intractable, nullifying any direct approach to estimate $\Sigma$. In §4 we indicate a direct bootstrap-based approach to perform valid inference.

## 4 STRONG INVARIANCE PRINCIPLE

Moving beyond the asymptotic normality of the Q-iterates, the primary goal of this section is to further deepen the understanding of their stochastic dynamics and to better characterize the asymptotic distributional approximation of the associated partial sum process by deriving a powerful probabilistic tool known as the *strong invariance principle*. Due to space constraints, we include a broad discussion on the relevant literature in §8. Due to the non-stationary nature of the sequence $(\mathbf{Q}_{t,n})_{t\geq1}$, its stochastic dynamics cannot be well captured by the standard Brownian process. Motivated by Bonnerjee et al. (2024), we instead propose approximating the partial sum process of $(\mathbf{Q}_{t,n})$ by that of a non-stationary Gaussian process specifically designed for matching the covariance structure. Specifically, let $\aleph_1, \ldots, \aleph_n \in \mathbb{R}^D$ be i.i.d. centered Gaussian random vectors with covariance matrix $\mathrm{Cov}(\aleph_t) = \mathrm{Cov}(Z_t)$. Then, in light of (2.1) and the linear approximation in (7.18), we define the Gaussian process $(Y_t)_{t\geq1}$ via $Y_0 = \mathbf{0}$ and

$$Y_t = (I - \eta_{t,n} G) Y_{t-1} + \eta_{t,n} \aleph_t, \quad t \geq 1, \quad (4.1)$$

where $G = I - \gamma H^{\pi_\star} \in \mathbb{R}^{D \times D}$. Throughout this section, we focus on the LD2Z schedule.

**Theorem 4.1.** *Grant Assumptions 3.1 and 3.2 for the MDP. Consider the learning rate PD2Z-$\nu$ learning rate and grant the assumptions of Theorem 3.5. Then, for all sufficiently large $n$, there exists a probability space on which one can define random vectors $\mathbf{Q}_1^c, \ldots, \mathbf{Q}_n^c$ such that $(\mathbf{Q}_{t,n}^c)_{t=1}^n \overset{\mathcal{D}}{=} (\mathbf{Q}_{t,n})_{t=1}^n$ and*

$$\max_{k_n \leq t \leq n} \left| \sum_{l=t}^{n} (\mathbf{Q}_l^c - \mathbf{Q}^\star - Y_l) \right|_\infty = o_{\mathbb{P}}(n^{\frac{1}{p} + \frac{1}{\nu+1}}),$$

*where $k_n = n - \lfloor cn^{\frac{\nu}{\nu+1}} \rfloor + 1$, and $c > 0$, $\nu > 1/p$ are constants.*

*Remark* 4.1. Theorem 4.1 provides the first strong Gaussian approximation for the partial sum process of Q-iterates with PD2Z-$\nu$ schedule. In the context of Q-learning, only functional central limit theorem is established Li et al. (2023b) for the polynomially decaying step sizes. A similar time-uniform approximation can also be established for the polynomially decaying learning schedule, which may be of independent interest.

**Theorem 4.2.** *Grant Assumptions 3.1 and 3.2 for the MDP. Consider the learning rate $\tilde{\eta}_t = \eta t^{-\beta}$ in (2.1) for $\eta > 0$, $\beta \in (1 - 1/p, 1)$, where $p$ is same as in Assumption 3.1. Then, there exists $(\aleph_t)_{t=1}^n \overset{i.i.d.}{\sim} N(0, \Gamma)$ such that, with*

$$\tilde{Y}_t = (I - \tilde{\eta}_t G)\tilde{Y}_{t-1} + \tilde{\eta}_t \aleph_t, Y_0 = \mathbf{0}, t \geq 1, \ G = I - \gamma H^{\pi_\star}, \tag{4.2}$$

*it holds that,*

$$\max_{1 \leq t \leq n} \left| \sum_{l=1}^t (\mathbf{Q}_l - \mathbf{Q}^\star - \tilde{Y}_l) \right|_\infty = o_\mathbb{P}(n^{1/p} \log n).$$

The key difference between the results of Theorems 4.1 and 4.2 is in the way partial sums are uniformly approximated. It is well-known that the polynomially decaying step-sizes offer attractive asymptotic properties; the optimality of Theorem 4.2, despite being new in the literature, is therefore not surprising. The strong approximation result is also classical in its expression, strongly echoing results such as Komlós et al. (1976). In fact, it can be argued that the approximation in Theorem 4.2 is much sharper than a functional CLT approximation Li et al. (2023b). As a toy example, consider the vanilla SGD setting, and suppose $K = 1$. Suppose $F(\theta) = (\theta - \mu)^2/2$, and $\nabla f(\theta, \xi) := \theta - \mu + \xi$. In this setting, the Gaussian approximation analogous to (4.2) is

$$Y_{t,n}^G = (I - \eta_{t,n} A)Y_{t-1,n}^G + \eta_{t,n} Z_t, \ Z_t \sim N(\mathbf{0}, \text{Var}(\xi)), \ Y_{0,n}^G = \mathbf{0}. \tag{4.3}$$

Here $A = \nabla_2 F(\mu) = I$. On the other hand, the vanilla SGD iterates can also be seen as $Y_{t,n} - \mu = (I - \eta_{t,n} A)(Y_{t-1,n} - \mu) + \eta_{t,n} \xi_t$. Therefore, it can be seen that $Y_{t,n} - \mu$ and $Y_{t,n}^G$ have exactly the same covariance structure, i.e. $\text{Cov}(Y_{s,n}^G, Y_{t,n}^G) = \text{Cov}(Y_{s,n}, Y_{t,n})$; on the other hand, even in such a simplified setting, an approximation by Brownian motion, such as that by functional CLT, captures the covariance structure of the iterates $\{Y_t - \mu\}_{t \geq 1}$ only in an asymptotic sense. The Gaussian approximation $Y_t^G$ in (4.3) is a particular example of covariance-matching approximations, introduced by Bonnerjee et al. (2024)- but generalized to account for the particular non-stationarity imposed by Q-learning iterates.

On the other hand, a strong approximation result for PD2Z-$\nu$ schedule works on the *tail* partial sums, much akin to the tail PR-averaged central limit theory. Moreover, the range of the approximation is also limited between $k_n$ and $n$, which may mean $n - \lfloor \sqrt{n} \rfloor$ to $n$ for the particular case of LD2Z schedule. Noticeably, despite the much faster decay from the initialization, for larger values of $\nu$, PD2Z-$\nu$ can also maintain a time-uniform strong approximation for almost the entire range of its steps. Moreover, in polynomially decaying step-sizes, in aiming for the optimality of strong invariance principles, the choice of $\beta \approx 1$ implies that the decay of $\mathbf{Q}_t$ from the initialization $\mathbf{Q}_0$ is $O(1)$; i.e. there is practically or very slow decay, which results in extremely slow convergence to the asymptotic regime. In contrast, even when uniform Gaussian approximation is assured, the inherent properties of the PD2Z-$\nu$ schedules do not affect convergence. Finally, no functional central limit theory is even known for these learning schedules.

Finally, we remark that as an immediate result of Theorem 4.1, for $p > 2$,

$$\sup_{z \geq 0} \left| \mathbb{P}\left( \max_{k_n \leq t \leq n} \left| \sum_{l=t}^n (\mathbf{Q}_l^c - \mathbf{Q}^\star) \right|_\infty \leq z \right) - \mathbb{P}\left( \max_{k_n \leq t \leq n} \left| \sum_{l=t}^n Y_l \right|_\infty \leq z \right) \right| \to 0. \tag{4.4}$$

Beyond theoretical interest, (4.4) hints at practical, bootstrap-based algorithms for time-uniform inference. In particular, the estimation of covariance matrix of $\mathbf{Q}_n$, especially for the PD2Z-$\nu$ learning schedule, may be significantly non-trivial. However, estimation of $\Gamma$ and $H^{\pi^*}$ can be essentially done using (2.2) and the fact that $B\mathbf{Q}^\star = \mathbf{Q}^\star$. This hints at an easily implementable Gaussian bootstrap procedure by running multiple independent chains of $Y_t$ parallelly. Similar inferential procedures have been proposed in a time-series context in Wu & Zhao (2007), and also more recently in Bonnerjee et al. (2025) in a local SGD setting.

## 5 SIMULATION RESULTS

In this section, we present some numerical experiments that empirically explore our theoretical results. In §5.2, we compare the performance of LD2Z schedule with the polynomially decaying and the constant learning rates, as well as the PD2Z-$\nu$ learning rates with $\nu = 2, 3$. Moving on, In §5.3 we investigate the accuracy of our time-uniform approximations. We also provide some additional simulation studies involving the central limit theorem in Appendix §5.4.

### 5.1 SET-UP

For each of the experiments, we consider a $4 \times 4$ gridworld with the slippery mechanism in Frozen-Lake (Zhang & Xie, 2024), and four actions (left/up/right/down). The discount factor is taken as $\gamma = 0.1$. There are two special states, $A$ and $B$, from which the agent can only intend to move to $A'$ and $B'$, respectively. Once an action is chosen according to the behavior policy, the agent moves in the intended direction with probability $0.9$, and with probability $0.05$ each, it instead moves in one of the two perpendicular directions. If the agent attempts to move outside the grid, it remains in the same state and receives a reward of $-1$. Otherwise, the reward depends on the current state, with $r(A) = 10, r(B) = 5$, and $r(s) = 0$ for all $s \neq A, B$.

### 5.2 COMPARATIVE PERFORMANCE BETWEEN LEARNING RATES.

In these experiments, we consider Q-learning with initialization at $\mathbf{0}$; since it's clearly evident in Figure 1 that LD2Z massively outperforms the polynomially decaying step size, we focus on LD2Z PD2Z-$\nu$ and constant learning schedules. For the experiments in Figure 2 (Left), we fix $n = 5000$, and run $B = 1000$ many Monte-Carlo Q-learning chains. Subsequently, for each learning schedules considered, we plot the mean error $|\mathbf{Q}_{t,n} - \mathbf{Q}^\star|_\infty$ for $1000 \leq t \leq n$ along with corresponding shaded bands indicating one standard deviation. On the other hand, for Figure 2 (Right), we run $B = 1000$ many independent Q-learning chains for each of $n \in \{500, 100, 1500, 2000, 2500\}$, and plot the mean error $|\mathbf{Q}_{n,n} - \mathbf{Q}^\star|_\infty$ against $n$, along with corresponding shaded bands.

Clearly the PD2Z-$\nu$ learning schedules outperforms the constant learning rate, which maintains a consistent bias having converged to a stationary distribution. On the other hand, increasing $\nu$ seems to have a small effect at reducing the error $|\mathbf{Q}_{t,n} - \mathbf{Q}^\star|_\infty$ when $t < n$. However, if we focus only on the final iterate error $|\mathbf{Q}_{n,n} - \mathbf{Q}^\star|_\infty$, the performance is similar across $\nu \in \{1, 2, 3\}$. This hints at a surprising stability across the PD2Z-$\nu$ class, justifying the widespread use of LD2Z schedule.

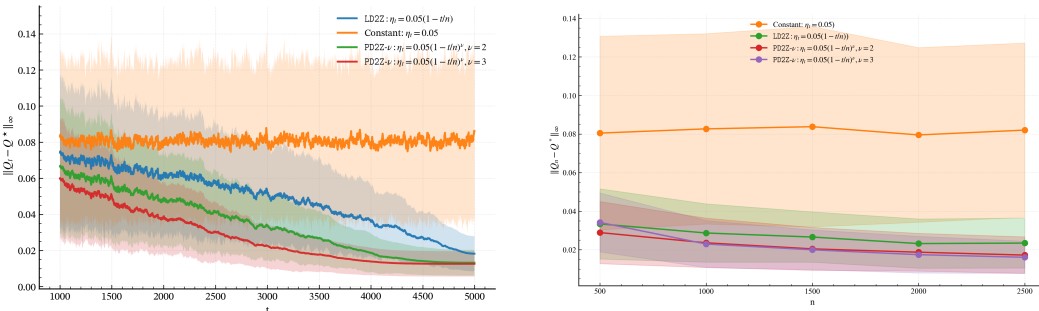

Figure 2: Performance comparison between LD2Z, PD2Z-$\nu$ with $\nu = 2, 3$ and constant learning schedules.

### 5.3 EXPERIMENTS ON TIME-UNIFORM APPROXIMATIONS.

In this section, we empirically investigate the time-uniform strong approximation results in Theorems 4.1 and 4.2. Working with the same $4 \times 4$ gridworld setting with number of iterations $n = 5000$ as in the previous section, in Figure 3 (Left), we consider the quantiles of $\max_{k_n \leq t \leq n} |\sum_{l=t}^n (\mathbf{Q}_l^c - \mathbf{Q}^\star)|_\infty$, for the LD2Z step-size $\eta_t = 0.05(1 - t/n)$ and compare them with the corresponding quantiles of $\max_{k_n \leq t \leq n} |\sum_{l=t}^n Y_l|_\infty$. All the quantiles are empirically calculated based on $B = 500$ Monte

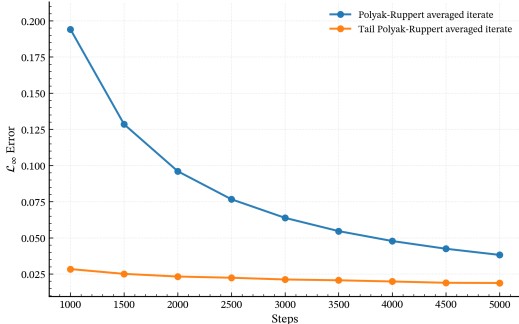

Figure 4: $\mathcal{L}_\infty$ error comparison of PR-averaged and tail PR-averaged iterates.

Carlo repetitions. Similarly, Figure 3 (Right) corresponds to the Gaussian approximation in Theorem 4.2 for the polynomially decaying learning rate $\eta_t = 0.05 t^{-0.65}$. In particular, Figure 3 (Right) also contains the corresponding quantiles of the Brownian motion based approximation (Theorem 3.1, Li et al. (2023b)). Despite the ubiquity of functional central limit theory, the sub-optimality of such approximation in terms of uniform approximation is evident. Together, these experiments establish the accuracy of the time-uniform approximations in §4, calling for their increased use in bootstrap procedures.

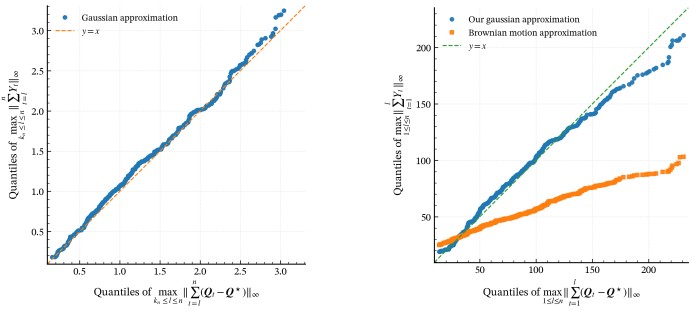

Figure 3: Q–Q plots of sup-norm distributions.

### 5.4 Central limit theory in practice.

This section is devoted to empirically validating the central limit theory established in §3.2. To that end, we first establish the efficacy of the *tail* Polyak-Ruppert averaged iterates $(\bar{\mathbf{Q}}_n)$ over the usual PR-averaged versions (denote by $\tilde{\mathbf{Q}}_n$) for LD2Z learning schedule. For $n \in \{1000, 1500, \ldots, 5000\}$, we estimate $\mathbb{E}[|\bar{\mathbf{Q}}_n - \mathbf{Q}^\star|_\infty]$ and $\mathbb{E}[|\tilde{\mathbf{Q}}_n - \mathbf{Q}^\star|_\infty]$ over $B = 1000$ Monte-Carlo repetitions. From the corresponding illustration in Figure 4, the superiority of $\bar{\mathbf{Q}}_n$ over $\tilde{\mathbf{Q}}_n$ is clear.

## 6 Discussion & Limitations

In this article, we develop asymptotic theory for the Q-learning with LD2Z and the more general PD2Z-$\nu$ learning schedules. Despite their increasing use in generative models, these learning schedules are yet to be thoroughly explored in the theoretical literature of stochastic approximation algorithms. To the best of our knowledge, this work constitutes the first one to include a systematic treatment of this step-size for Q-learning. Future extensions include the theory for the potential bootstrap algorithm and Berry-Esseen bounds to properly quantify the central limit theory.

Moreover, as pointed out by a reviewer, LD2Z step-size schedule is applicable primarily in offline reinforcement learning settings with pre-collected datasets, where the total sample size $n$ is known in advance. We acknowledge this as the main limitation of the LD2Z schedule when applied to Q-learning. However, our methods allow for the case where $n$ is mis-specified. Let $n_0 \leq n$ denote

the true sample size, while $n$ is used in the `LD2Z` step-size schedule. Then, as long as the mis-specification satisfies $n - n_0 \leq \alpha\sqrt{n}$ for some constant $\alpha \in (0, 1)$, our asymptotic results remain valid. Generalizing `LD2Z` and `PD2Z-`$\nu$ to online RL set-up constitutes an interesting direction, and warrants further research.

## ETHICS STATEMENT

The research follows all ethical guidelines. No human data or ethically sensitive content is involved. All potential limitations and justifications are adequately addressed. We do not anticipate any negative impacts, and as such the paper does not include a dedicated speculative discussion of broader societal impacts.

## REPRODUCIBILITY STATEMENT

All the relevant reproducible codes and figures can be found in the Github repository. All the theoretical results and assumptions are rigorously proved and validated in §7 and §8.

## AUTHOR CONTRIBUTIONS

All the authors contributed equally to this research.

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

## 7 APPENDIX A

In this section we collect the proofs of Theorems 3.1 and 3.5.

*Proof of Theorem 3.1.* Denote $\Delta_{t,n} := \mathbf{Q}_{t,n} - \mathbf{Q}^\star$. Then, it is immediate that

$$\begin{aligned}
\Delta_{t,n} &= (1 - \eta_{t,n})(\mathbf{Q}_{t-1,n} - \mathbf{Q}^\star) + \eta_{t,n}(\hat{B}_t\mathbf{Q}_{t-1,n} - B(\mathbf{Q}^\star)) \\
&= A_t\Delta_{t-1,n} + \eta_{t,n}Z_t + \gamma\eta_{t,n}(M_{t,n} + (H^{\pi_{t-1,n}} - H^{\pi^\star})\mathbf{Q}_{t-1,n}),
\end{aligned} \tag{7.1}$$

where $A_t = I - \eta_{t,n}G$, and $M_{t,n} = (\mathcal{P}_t - \mathcal{P})(V_{t-1,n} - V^\star)$. From the definition of greedy policy, it follows that $(H^{\pi_{t-1,n}} - H^{\pi^\star})\mathbf{Q}^\star \leq 0$, where $\leq$ and $\geq$ are interpreted element-wise. Therefore, clearly

$$\Delta_{t,n} \leq \left(I - \eta_{t,n}(I - \gamma H^{\pi_{t-1,n}})\right)\Delta_{t-1,n} + \eta_{t,n}(Z_t + \gamma M_{t,n}),$$

which directly yields, via Proposition 4 of that

$$\begin{aligned}
\|\Delta_{t,n}\|_p^2 &\leq (1 - \eta_{t,n}(1-\gamma))^2\|\Delta_{t-1,n}\|_p^2 + 2(p-1)\eta_{t,n}^2(\|Z_t\|_p^2 + \gamma^2\|M_{t,n}\|_p^2) \\
&\leq \left((1 - \eta_{t,n}(1-\gamma))^2 + 2(p-1)\eta_{t,n}^2\gamma^2\right)\mathbb{E}[|\Delta_{t-1,n}|^2] + \eta_{t,n}^2 c_p,
\end{aligned}$$

with $c_p = 2(p-1)\Theta_p^{2/p}$. Recursively, it holds that

$$\|\Delta_{t,n}\|_p^2 \leq \tilde{\mathcal{A}}_0^t|\Delta_0|^2 + c_p\sum_{s=1}^t \eta_{s,n}^2\tilde{\mathcal{A}}_s^t, \tag{7.2}$$

where $\tilde{\mathcal{A}}_s^t = \prod_{j=s+1}^t(1 - \eta_{j,n}c_1 + \eta_{j,n}^2 c_2)$, where $c_1 = 2(1-\gamma), c_2 = (1-\gamma)^2 + 2(p-1)\gamma^2$. From the choice of $\eta$ satisfying $\eta c_1 - \eta^2 c_2 > 0$, we can derive

$$\tilde{\mathcal{A}}_s^t \leq \mathcal{A}_s^t := \prod_{j=s+1}^t (1 - \eta_{j,n}c_3),$$

for some small constant $c_3 \in (0, 1)$. In light of $\sum_{j=1}^t \eta_{j,n} \geq \eta t(1 - n^{-1})^\nu$, we have $\mathcal{A}_0^t \leq \exp(-c_3\eta(1 - n^{-1})^\nu t)$. Therefore, applying Lemma 11.1 the proof is completed. $\square$

*Proof of Theorem 3.5.* We consider deriving the Gaussian approximation through a series of steps. In particular, our proof strategy is to linearize the Q-learning iterates before applying suitable, off-the-shelf central limit theory. The steps till linearization are not straightforward, especially in light of the complications arising out of PD2Z-$\nu$ learning rates. In particular, the non-linearity of the Bellman operator requires careful tempering. We provide the formal proof in the following. Throughout the proof, we let $k_n = n - \lfloor cn^{\frac{\nu}{\nu+1}} \rfloor$.

### 7.1 STEP I

Let $\mathbf{Q}_0^\diamond = \mathbf{Q}^\star$, and define the oracle Q-learning iterates

$$\mathbf{Q}_{t,n}^\diamond = (1 - \eta_{t,n})\mathbf{Q}_{t-1,n}^\diamond + \eta_{t,n}\hat{B}_t\mathbf{Q}_{t-1,n}^\diamond, \ t \geq 1. \tag{7.3}$$

Note that

$$\begin{aligned}
|\mathbf{Q}_{t,n} - \mathbf{Q}_{t,n}^\diamond|_\infty &\leq (1 - \eta_{t,n})|\mathbf{Q}_{t-1,n} - \mathbf{Q}_{t-1,n}^\diamond|_\infty + \eta_{t,n}|\hat{B}_t\mathbf{Q}_{t,n} - \hat{B}_t\mathbf{Q}_{t,n}^\diamond|_\infty \\
&\leq (1 - \eta_{t,n}(1-\gamma))|\mathbf{Q}_{t-1,n} - \mathbf{Q}_{t-1,n}^\diamond|_\infty \\
&\quad\vdots \\
&\leq Y_0^t(1-\gamma)\,|\mathbf{Q}_0 - \mathbf{Q}^\star|_\infty, \tag{7.4}
\end{aligned}$$

where for $c > 0$, $Y_i^t(c) = \prod_{j=i+1}^t(1 - \eta_{j,n}c)$, and the second inequality in (7.4) follows from the contraction of Bellman operators (2.2). Elementary calculations show that $Y_0^t(1-\gamma) \lesssim_\gamma$

$\exp(-c_{\nu,\gamma,\eta}t)$ for some $c > 0$, which implies, via (7.4), that

$$n^{\frac{\nu}{2(\nu+1)}}|\bar{\mathbf{Q}}_n - \bar{\mathbf{Q}}_n^\diamond| \leq n^{\frac{\nu}{2(\nu+1)}}(n-k_n)^{-1}\sum_{t=k_n}^{n}|\mathbf{Q}_{t,n} - \mathbf{Q}_{t,n}^\diamond|_\infty$$

$$\lesssim n^{-\frac{\nu}{2(\nu+1)}}\int_1^n \exp(-ct)\,\mathrm{d}t$$

$$= O(n^{-\frac{\nu}{2(\nu+1)}}) \text{ almost surely.} \tag{7.5}$$

Therefore, Step I enables us to investigate the asymptotic properties of $\bar{\mathbf{Q}}^\diamond$.

## 7.2 STEP II

Define the empirical version of $\mathcal{P}$ as

$$\mathcal{P}_t((s,a),\cdot) = (\mathbf{1}_{s_t=s',s_{t-1}=s,a_{t-1}=a})_{s'\in S}. \tag{7.6}$$

In other words, $\mathcal{P}_t \in \mathbb{R}^{D\times|S|}$ is a matrix with one-hot-coded rows. Moreover, let

$$V_{t,n}(s) = \max_{a'\in\mathcal{A}}\mathbf{Q}_{t,n}(s,a), \text{ and } V^\star(s) = \max_{a'\in\mathcal{A}}\mathbf{Q}^\star(s,a), \tag{7.7}$$

with $V_{t,n} = (V_{t,n}(s))_{s\in\mathcal{S}} \in \mathbb{R}^{|\mathcal{S}|}$, and $V^\star$ likewise defined. Note that,

$$\mathcal{P}_t V_{t-1,n} = \max_{a'\in\mathcal{A}}\mathbf{Q}_{t-1,n}\left(N\left(s,a,U_t\right),a'\right),$$

and $\mathcal{P}V_{t-1,n} = \mathbb{E}[\mathcal{P}_t V_{t-1,n}|\mathcal{F}_{t-1}]$ where $\mathcal{F}_{t-1}$ is the $\sigma$-field induced by the random variables $(U_s,V_s)_{s\leq t}$. Clearly, $\mathcal{P}V^\star = \mathbb{E}[\max_{a'\in\mathcal{A}}\mathbf{Q}^\star\left(N\left(s,a,U\right),a'\right)]$, $U \sim U[0,1]$. Observe that

$$\hat{B}_t\mathbf{Q}_{t-1,n}^\diamond - B\mathbf{Q}^\star = \hat{B}_t\mathbf{Q}_{t-1,n}^\diamond - \hat{B}_t\mathbf{Q}^\star + Z_t \tag{7.8}$$

$$= \gamma\mathcal{P}_t(V_{t-1,n} - V^\star) + Z_t \tag{7.9}$$

$$= \gamma\left(M_{t,n} + (H^{\pi_{t-1,n}^\diamond} - H^{\pi^\star})\mathbf{Q}_{t-1,n}^\diamond + \gamma H^{\pi^\star}(\mathbf{Q}_{t-1,n}^\diamond - \mathbf{Q}^\star)\right) + Z_t, \tag{7.10}$$

where (7.8) follows from $Z_t = \hat{B}_t\mathbf{Q}^\star - B\mathbf{Q}^\star$; (7.9) is implied by (2.2), and (7.10) is obtained after defining $M_{t,n} = (\mathcal{P}_t - \mathcal{P})(V_{t-1,n} - V^\star)$. Note that, in particular $Z_t$ are mean-zero i.i.d. random variables, and $(M_{t,n})_{t\geq 1}$ is a martingale difference sequence. Now, using $B(\mathbf{Q}^\star) = \mathbf{Q}^\star$ and (7.8)-(7.10), rewrite (7.3) as

$$\Delta_{t,n} := \mathbf{Q}_{t,n}^\diamond - \mathbf{Q}^\star = (1-\eta_{t,n})(\mathbf{Q}_{t-1,n}^\diamond - \mathbf{Q}^\star) + \eta_{t,n}(\hat{B}_t\mathbf{Q}_{t-1,n}^\diamond - B(\mathbf{Q}^\star))$$

$$= A_t\Delta_{t-1,n} + \eta_{t,n}Z_t + \gamma\eta_{t,n}(M_{t,n} + (H^{\pi_{t-1,n}^\diamond} - H^{\pi^\star})\mathbf{Q}_{t-1,n}^\diamond), \tag{7.11}$$

where $A_{t,n} = I - \eta_{t,n}G$, $G = I - \gamma H^{\pi^\star}$, and $\Delta_0 = \mathbf{0}$. Define another "sandwich" sequence as follows:

$$\Delta_{t,n}^{(L)} = A_t\Delta_{t-1,n}^{(L)} + \eta_{t,n}Z_t + \gamma\eta_{t,n}M_{t,n}, \ \Delta_0^{(2)} = \mathbf{0}. \tag{7.12}$$

Following the property of optimal policy, it is immediate that $(H^{\pi_t^\diamond} - H^{\pi^\star})\mathbf{Q}_{t-1,n} \geq 0$, and hence,

$$\Delta_{t,n}^{(L)} \leq \Delta_{t,n}. \tag{7.13}$$

Moreover, it follows that

$$\mathbb{E}[|\Delta_{t,n} - \Delta_{t,n}^{(L)}|_\infty] \leq (1-\eta_{t,n}(1-\gamma))\mathbb{E}[|\Delta_{t-1,n} - \Delta_{t-1,n}^{(L)}|_\infty] + \mathbb{E}[|(H^{\pi_{t-1,n}^\diamond} - H^{\pi^\star})\mathbf{Q}_{t-1,n}^\diamond|_\infty]$$

$$\leq (1-\eta_{t,n}(1-\gamma))\mathbb{E}[|\Delta_{t-1,n} - \Delta_{t-1,n}^{(L)}|_\infty] + \gamma\eta_{t,n}\mathbb{E}[|(H^{\pi_{t-1,n}^\diamond} - H^{\pi^\star})\Delta_{t-1,n}|_\infty] \tag{7.14}$$

$$\leq (1-\eta_{t,n}(1-\gamma))\mathbb{E}[|\Delta_{t-1,n} - \Delta_{t-1,n}^{(L)}|_\infty] + \gamma L\eta_{t,n}\mathbb{E}[|\Delta_{t-1,n}|_\infty^2] \tag{7.15}$$

$$= \gamma L\sum_{s=0}^{t}\eta_{s,n}\mathcal{A}_s^t\mathbb{E}[|\mathbf{Q}_{s,n} - \mathbf{Q}^\star|_\infty^2]$$

$$\lesssim \sum_{s=0}^{k_n}\eta_{s,n}^2\mathcal{A}_s^t + n^{-\frac{\nu}{\nu+1}}\sum_{s=k_n+1}^{t}\eta_{s,n}\mathcal{A}_s^t \lesssim n^{-\frac{\nu}{\nu+1}}. \tag{7.16}$$

where (7.14) follows from noting that $(H^{\pi^\diamond_{t-1,n}} - H^{\pi^\star})\mathbf{Q}^\star \leq 0$; (7.15) follows from Assumption 3.2, and (7.16) involves an application of Theorem 3.1 and Lemma 11.1. Clearly, (7.16) produces

$$n^{\frac{\nu}{2(\nu+1)}} \mathbb{E}[|\bar{\Delta}_n - \bar{\Delta}_n^{(L)}|_\infty] = O(n^{-\frac{\nu}{2(\nu+1)}})$$

which implies that

$$n^{\frac{\nu}{2(\nu+1)}} \left(\bar{\Delta}_n - \bar{\Delta}_n^{(L)}\right) \xrightarrow{\mathbb{P}} 0. \tag{7.17}$$

## 7.3 STEP III

In this step, we will show that both $\Delta_{t,n}^{(L)}$ is well-approximated by a linear process. To that end, further define

$$X_{t,n} = A_t X_{t-1,n} + \eta_{t,n} Z_t, \; X_0 = \mathbf{0}. \tag{7.18}$$

With this definition established, we can proceed to approximate $\Delta_{t,n}^{(L)}$ by $X_{t,n}$. Indeed, with $\Delta_{t,n}^{(L)} := \Delta_{t,n}^{(L)} - X_{t,n} \in \mathbb{R}^D$.

$$\begin{aligned}
\mathbb{E}[|\Delta_{t,n}^{(L)}|_\infty^2] &\lesssim_D \mathbb{E}[|\Delta_{t,n}^{(L)}|_2^2] = \mathbb{E}[|(I - \eta_{t,n}(I - \gamma H^{\pi^\star}))\Delta_{t-1,n}^{(L)}|_2^2] + \gamma \eta_{t,n}^2 \mathbb{E}[|M_{t,n}|_2^2] \\
&\leq (1 - \eta_{t,n}(1-\gamma))\mathbb{E}[|\Delta_{t-1,n}^{(L)}|_2^2] + \gamma \eta_{t,n}^2 2\mathbb{E}[|V_{t-1,n} - V^\star|_2^2] \\
&\lesssim \sum_{s=1}^{t} \eta_{s,n}^2 \mathcal{A}_s^t \mathbb{E}[|V_{s-1} - V^\star|^2] \\
&\lesssim \sum_{s=0}^{k_n} \eta_{s,n}^3 \mathcal{A}_s^t + n^{-\frac{\nu}{\nu+1}} \sum_{s=k_n+1}^{t} \eta_{s,n}^2 \mathcal{A}_s^t \lesssim n^{-2\frac{\nu}{\nu+1}}
\end{aligned} \tag{7.19}$$

where the second equality uses the fact that $M_{t,n}$ are martingale differences; the inequality in the third assertion involves (i) using that $H^{\pi^\diamond_{t-1,n}}$ is a stochastic matrix to deduce $|I - \eta_{t,n}(I - \gamma H^{\pi^\star})|_\infty = 1 - \eta_{t,n}(1-\gamma)$, and (ii) using that both $\mathcal{P}_t$ and $\mathcal{P}$ are stochastic matrices to obtain $|P_t - P|_\infty \leq 2$; the final assertion invokes Theorem 3.1 and Lemma 11.1. Equation 7.19 immediately results in

$$n^{\frac{\nu}{2(\nu+1)}} \mathbb{E}[|\bar{\Delta}_n^{(L)} - \bar{X}_n|_\infty] = n^{-\frac{\nu}{2(\nu+1)}} \sum_{t=k_n}^{n} \sqrt{\mathbb{E}[|\Delta_{t,n}^{(L)}|_\infty^2]} = O(n^{-\frac{\nu}{2(\nu+1)}}),$$

which, similar to (7.17) implies that

$$n^{\frac{\nu}{2(\nu+1)}} \left(\bar{\Delta}_n^{(L)} - \bar{X}_n\right) \xrightarrow{\mathbb{P}} 0. \tag{7.20}$$

## 7.4 STEP IV

In light of (7.5), (7.17) and (8.3), the proof is complete if one derives a central limit theory of $\bar{X}_n = (n - k_n)^{-1} \sum_{t=k_n}^{n} X_{t,n}$. To that end, re-write

$$\sum_{t=k_n}^{n} X_{t,n} = \sum_{s=1}^{n} \eta_{s,n} \mathcal{V}_{s,n} Z_s, \; \mathcal{V}_{s,n} = \sum_{t=s\vee k_n}^{n} \mathbf{A}_{s,n}^t$$

where $\mathbf{A}_{s,n}^t = \prod_{j=s+1}^{t} A_{j,n}$. We proceed step-by-step. Let $L_{s,n} = s \vee k_n$. Firstly, note that

$$\sum_{s=1}^{n} \eta_{s,n}^2 |\mathcal{V}_{s,n}|_F^2 \lesssim n^{\frac{\nu}{\nu+1}} \sum_{s=1}^{n} \sum_{t=L_{s,n}+1}^{n} \eta_{s,n}^2 |\mathbf{A}_{s,n}^t|_F^2 \leq n^{\frac{\nu}{\nu+1}} \sum_{t=k_n}^{n} \sum_{s=1}^{t} \eta_{s,n}^2 |\mathbf{A}_{s,n}^t|_F^2 = O\left(n^{\frac{\nu}{\nu+1}} \frac{\sum_{t=k_n}^{n}(n-t)^\nu}{n^\nu}\right)$$
$$= O(n^{\frac{\nu}{\nu+1}}), \tag{7.21}$$

which establishes the Lindeberg condition that $n^{-\frac{\nu}{2(\nu+1)}} \max_s \eta_{s,n}|\mathcal{V}_{s,n}| = O(1)$. Now we shift focus to showing that

$$W_n := n^{-\frac{\nu}{\nu+1}} \sum_{s=1}^{n} \eta_{s,n}^2 \mathcal{V}_{s,n} \Gamma \mathcal{V}_{s,n}^\top \to \Sigma$$

for some $\Sigma \succ 0$. Write

$$W_n = (1 - 1/n)^{\frac{\nu}{\nu+1}} W_{n-1} + R_n,$$

where

$$R_n := n^{-\frac{\nu}{\nu+1}} \sum_{s=1}^{n-1} \left[ (C_{s,n} - C_{s,n-1}) \Gamma C_{s,n-1}^\top + C_{s,n} \Gamma (C_{s,n} - C_{s,n-1})^\top \right], \ C_{s,n} = \eta_{s,n} \mathcal{V}_{s,n}.$$

(7.22)

The proof follows by showing that $nR_n$ is a Cauchy sequence in $\mathbb{R}^{d\times d}$ through an argument mimicking Lemma 11.1, and we omit the details for brevity. Finally, our conclusion follows from (7.21) via Lindeberg-Feller central limit theory. $\qquad\square$

# 8 APPENDIX B: DISCUSSION ON STRONG APPROXIMATION OF Q-LEARNING ITERATES

**Related Literature.** The method of *invariance principle* was introduced by Erdös & Kac (1946) and has since been extensively studied, serving as a powerful tool for analyzing distributional properties in a wide range of statistical inference problems (Csörgö & Hall, 1984; Csörgo & Révész, 2014). Applications include nonparametric simultaneous inference (Liu & Wu, 2010; Karmakar et al., 2022), change-point detection and inference (Wu & Zhao, 2007), online statistical inference (Lee et al., 2022; Zhu et al., 2024; Li et al., 2023b), and construction of time-uniform confidence sequences (Waudby-Smith et al., 2024; Xie et al., 2024).

For independent and identically distributed (i.i.d.) random variables, Strassen (1964) initialed the study of almost sure approximation for the partial sums by Wiener process, and was later refined by Csörgő & Révész (1975a) and Csörgő & Révész (1975b). The optimal strong approximation in this setting was established in the celebrated work (Komlós et al., 1975; 1976). Specifically, let $\xi_1, \ldots, \xi_n \in \mathbb{R}$ be i.i.d. centered random variables with $\mathrm{Var}(\xi_1) = \sigma^2$ and $\mathbb{E}|\xi_1|^p < \infty$ for some constant $p > 2$. Then, for the sequence of partial sums $\{S_t\}_{t=1}^n$, where $S_t = \sum_{j=1}^t \xi_j$, there exists a probability space on which one can define random variables $\xi_1^c, \ldots, \xi_n^c$ with the partial sum process $S_t^c = \sum_{j=1}^t \xi_j^c, t \geq 1$, and a Brownian motion $\mathbb{B}(\cdot)$ such that $\{S_t^c\}_{t=1}^n \overset{\mathcal{D}}{=} \{S_t\}_{t=1}^n$ and

$$\max_{1 \leq t \leq n} |S_t^c - \sigma \mathbb{B}(t)| = o_{a.s.}(n^{1/p}).$$

Extensions of this result to multidimensional independent (but not necessarily identically distributed) random vectors has been developed by Einmahl (1987), Shao (1995), Götze & Zaitsev (2009), among others. Another line of research, more relevant to the online learning where the outputs may exhibit temporal dependence, has focused on generalizing the above strong approximation to dependent data; see, for example, Heyde & Scott (1973), Lu & Shao (1987), Wu (2007), Liu & Lin (2009), Dedecker et al. (2012), Merlevède & Rio (2012), among others. A notable contribution in this direction was made by Berkes et al. (2014), who established the optimal strong approximation for a broad class of causal stationary sequence $\{\xi_t\}_{t\geq 1}$. Under mild regularity conditions, they proved that

$$\max_{1 \leq t \leq n} |S_t^c - \sigma_\infty \mathbb{B}(t)| = o_{a.s.}(n^{1/p}),$$

(8.1)

where $\sigma_\infty^2 = \sum_{t\in\mathbb{Z}} \mathrm{Cov}(\xi_0, \xi_t) = \lim_{n\to\infty} \mathrm{Var}(S_n)/n$ stands for the long-run variance. This result implies that the process $\{\sigma_\infty \mathbb{B}(t)\}_{t=1}^n$ can preserve the second-order properties of $\{S_t\}_{t\geq 1}$ asymptotically.

However, in the context of Q-learning with time-varying step sizes, these results do not apply due to the nonstationary nature of the iterates $\{\mathbf{Q}_{t,n}\}_{t\geq 1}$ defined in (2.1). Unfortunately, strong approximations for non-stationary data remain relatively underexplored. Some contributions include Wu & Zhou (2011), Karmakar & Wu (2020) and Mies & Steland (2023), which lead to the following result: there exists a Gaussian process $\{\mathcal{G}_t\}_{t\geq 1}$ such that $\mathrm{Cov}(\mathcal{G}_t, \mathcal{G}_s) \approx \mathrm{Cov}(S_t, S_s)$ and

$$\max_{1 \leq t \leq n} |S_t^c - \mathcal{G}_t| = o_{\mathbb{P}}(\tau_n).$$

(8.2)

Compared to $\{\sigma_\infty \mathbb{B}(t)\}$ in (8.1), this more general $\{\mathcal{G}_t\}$ can better capture the dependence structure of $\{S_t\}$, as it allows potentially non-stationary increments $\{\mathcal{G}_t - \mathcal{G}_{t-1}\}_{t\geq 1}$. However, until the recent

work of Bonnerjee et al. (2024), it remained unclear how to explicitly construct such a process with optimal convergence rate. They provided an optimal Gaussian approximation of the form (8.2) with optimal $\tau_n = n^{1/p}$ and an explicit construction of the coupling Gaussian process $\{\mathcal{G}_t\}$. Motivated by this, one of the main objectives of this paper is to derive an optimal Gaussian approximation for $Q$-learning, including an explicit construction of the coupling Gaussian process. It is important to note that the dependence structure of $\{\mathbf{Q}_{t,n}\}_{t \geq 1}$ is significantly more complex than that considered in Bonnerjee et al. (2024), and thus their results are not directly applicable.

Now we proceed to the proofs of the results in §4.

*Proof of Theorem 4.1.* From equations (7.4), (7.16) and (7.19) it also follows that

$$\max_{k_n \leq t \leq n} | \sum_{s=t}^{n} (\mathbf{Q}_{s,n} - \mathbf{Q}^\star - X_{s,n})| = O_{\mathbb{P}}(1). \tag{8.3}$$

Note that (7.18) can be cast into the following form:

$$X_{t,n} = \sum_{s=1}^{t} \eta_s \mathbf{A}_{s-1,n}^t Z_s, \tag{8.4}$$

where $\mathbf{A}_{s,n}^t = \prod_{j=s+1}^{t} A_{j,n}$, $s, t \geq 0$, and $\mathbf{A}_t^t := I$ for $t \geq 1$. Moreover, using Theorem 4 of Götze & Zaitsev (2009), on a possibly enriched probability space, there exists $\aleph_t \overset{i.i.d.}{\sim} N(0, \Gamma)$, such that

$$\max_{1 \leq t \leq n} | \sum_{s=1}^{t} (Z_s - \aleph_s)|_\infty = o_{\mathbb{P}}(n^{1/p}). \tag{8.5}$$

If one defines $Y_t$ as

$$Y_t = (I - \eta_{t,n}(I - \gamma H^{\pi^\star}))Y_{t-1} + \eta_{t,n}\aleph_t,$$

then, for $t \geq k_n$,

$$\begin{aligned}
\sum_{l=t}^{n}(X_l - Y_l) &= \sum_{l=t}^{n} \sum_{s=1}^{l} \eta_{s,n} \mathbf{A}_{s-1,n}^l (Z_s - \aleph_s) \\
&= \sum_{s=1}^{n} \sum_{l=s \vee t}^{n} \eta_{s,n} \mathbf{A}_{s-1,n}^l (Z_s - \aleph_s) \\
&= \sum_{s=1}^{t} \sum_{l=t}^{n} \eta_{s,n} \mathbf{A}_{s-1,n}^l (Z_s - \aleph_s) + \sum_{s=t+1}^{n} \sum_{l=s}^{n} \eta_{s,n} \mathbf{A}_{s-1,n}^l (Z_s - \aleph_s). \tag{8.6}
\end{aligned}$$

Let us tackle the terms in (8.6) one-by-one. In particular, a similar treatment as Lemma 11.1 provides that for all $s \in [n]$

$$\max_{k_n \leq t \leq n} \max_{1 \leq s \leq t} \eta_{s,n} \sum_{l=t}^{n} |\mathbf{A}_{s-1,n}^l|_F = O(1).$$

Denote $\Gamma_{s,n}^t = \eta_{s,n} \sum_{l=t}^{n} A_{s-1,n}^l$. Therefore, for the first term in (8.6), one obtains

$$\begin{aligned}
&\max_{k_n \leq t \leq n} | \sum_{s=1}^{t} \sum_{l=t}^{n} \eta_{s,n} \mathbf{A}_{s-1,n}^l (Z_s - \aleph_s)|_\infty \\
&\leq \max_{k_n \leq t \leq n} \left( |\Gamma_{1,n}^t|_F + \sum_{s=2}^{t} |\Gamma_{s,n}^t - \Gamma_{s-1,n}^t|_F \right) \max_{k_n \leq t \leq n} | \sum_{s=1}^{t} (Z_s - \aleph_s)|_\infty \\
&= o_{\mathbb{P}}(n^{\frac{1}{p} + \frac{1}{\nu+1}}), \tag{8.7}
\end{aligned}$$

where the $o_{\mathbb{P}}$ assertion follows from (8.5), and the $n^{\frac{1}{\nu+1}}$ rate appears due to

$$\Gamma_{s,n}^t - \Gamma_{s-1,n}^t = \left( \frac{\eta_{s,n} - \eta_{s-1,n}}{\eta_{s,n}} I - \frac{\eta_{s-1,n}^2}{\eta_{s,n}} A \right) \Gamma_{s-1,n}^t,$$

upon which we invoke Lemma 11.1 along with $t \geq k_n$, and the elementary bound $\max_t \sum_{s=1}^{t} \frac{\eta_{s,n} - \eta_{s-1,n}}{\eta_{s,n}} \lesssim \log n$. The assertion for the second term follows from noting

$$\max_{k_n \leq t \leq n} \max_{t \leq s \leq n} \eta_{s,n} \sum_{l=s}^{n} |\mathbf{A}_{s-1,n}^l|_F \leq \max_{k_n \leq t \leq n} \max_{1 \leq s \leq t} \eta_{s,n} \sum_{l=t}^{n} |\mathbf{A}_{s-1,n}^l|_F,$$

after which we follow a similar argument as above. This completes the proof. $\qquad \square$

*Proof of Theorem 4.2.* We follow a proof similar to that of Theorem 4.1. Since the learning rates no longer depend on the number of iterations $n$, we omit the $n$ from the subscript.

## 8.1 STEP I

Similar to Step I in Theorem 4.2, elementary calculations show that $Y_0^t(1-\gamma) \lesssim_\gamma \exp(-ct^{1-\beta})$ for some $c > 0$, which implies, via (7.4), that

$$\max_{1 \leq t \leq n} |\sum_{s=1}^{t}(\mathbf{Q}_s - \mathbf{Q}_s^\diamond)|_\infty \leq \sum_{t=1}^{n}|\mathbf{Q}_t - \mathbf{Q}_t^\diamond|_\infty \lesssim \int_1^n \exp(-ct^{1-\beta}) = O(1) \text{ almost surely.} \quad (8.8)$$

## 8.2 STEP II

In this case, it follows that

$$\begin{aligned}
\mathbb{E}[|\Delta_t - \Delta_t^{(L)}|_\infty] &\leq (1 - \eta_t(1-\gamma))\mathbb{E}[|\Delta_{t-1} - \Delta_{t-1}^{(L)}|_\infty] + \mathbb{E}[|(H^{\pi_{t-1}^\diamond} - H^{\pi^\star})\mathbf{Q}_{t-1}^\diamond|_\infty] \\
&\leq (1 - \eta_t(1-\gamma))\mathbb{E}[|\Delta_{t-1} - \Delta_{t-1}^{(L)}|_\infty] + \gamma\eta_t\mathbb{E}[|(H^{\pi_{t-1}^\diamond} - H^{\pi^\star})\Delta_{t-1}|_\infty] \\
&\leq (1 - \eta_t(1-\gamma))\mathbb{E}[|\Delta_{t-1} - \Delta_{t-1}^{(L)}|_\infty] + \gamma L\eta_t\mathbb{E}[|\Delta_{t-1}|_\infty^2] \\
&\leq (1 - \eta_t(1-\gamma))\mathbb{E}[|\Delta_{t-1} - \Delta_{t-1}^{(L)}|_\infty] + L^2 C\eta_t^2, \quad (8.9)
\end{aligned}$$

where (8.9) involves an application of Theorem E.2 of Li et al. (2023b). Clearly, in lieu of $\beta > 1 - 1/p$, (8.9) entails

$$\mathbb{E}[|\Delta_t - \Delta_t^{(L)}|_\infty] = O(\eta_t),$$

which produces

$$\max_{1 \leq t \leq n} |\sum_{s=1}^{t}(\Delta_t - \Delta_t^{(L)})|_\infty = o_\mathbb{P}(n^{1/p}). \quad (8.10)$$

## 8.3 STEP III

In this step, we have,

$$\begin{aligned}
\mathbb{E}[|\delta_t^{(L)}|_\infty^2] \lesssim_D \mathbb{E}[|\delta_t^{(L)}|_2^2] &= \mathbb{E}[|(I - \eta_t(I - \gamma H^{\pi^\star}))\delta_{t-1}^{(L)}|_2^2] + \gamma\eta_t^2\mathbb{E}[|M_t|_2^2] \\
&\leq (1 - \eta_t(1-\gamma))\mathbb{E}[|\delta_{t-1}^{(L)}|_2^2] + \gamma\eta_t^2 2\mathbb{E}[|V_{t-1} - V^\star|_2^2] \\
&\leq (1 - \eta_t(1-\gamma))\mathbb{E}[|\delta_{t-1}^{(L)}|_2^2] + O(\eta_t^3), \quad (8.11)
\end{aligned}$$

whereupon one invokes Theorem E.2 of Li et al. (2023b) to conclude $\mathbb{E}[|\Delta_{t-1}|_\infty^2] = O(\eta_t)$. Equation (8.11) immediately results in

$$\max_{1 \leq t \leq n} |\sum_{s=1}^{t}(\Delta_t^{(L)} - X_t)| = O_\mathbb{P}(n^{1-\beta}) = o_\mathbb{P}(n^{1/p}), \quad (8.12)$$

similar to (8.10).

### 8.4 STEP IV

This step also follows similar to that of Theorem 4.1 by denoting $B_{s,n} = \eta_s \sum_{j=s}^{n} \mathbf{A}_{s-1}^{j}$ and observing

$$\max_{1 \leq t \leq n} |\sum_{s=1}^{t}(X_t - Y_t)|_\infty \leq \max_{1 \leq t \leq n} \Omega_t |\sum_{s=1}^{t}(Z_s - \aleph_s)|_\infty = o_{\mathbb{P}}(n^{1/p}\log n), \qquad (8.13)$$

where

$$\Omega_t = |B_1, t|_\infty + \sum_{s=2}^{t}|B_{s,t} - B_{s-1,t}|_\infty,$$

and the $\log n$ comes from Proposition 2 of Bonnerjee et al. (2025). Note that by construction, $(X_t^c)_{t\geq 1} \overset{d}{=} (X_t)_{t\geq 1}$. The proof is concluded by combining (8.8), (8.10), (8.12) and (8.13). $\qquad\square$

## 9 APPENDIX C: DERIVATION OF ASSUMPTION 3.2

The key insight behind the Assumption 3.2 is ensuring that the optimal policy, and the optimal quality function is unique. In that regard, we consider the following simple margin condition, that can be more illuminating in the Q-learning context.

**Assumption 9.1.** *The greedy policy* $\pi^\star(s) = \arg\max_a Q^\star(s,a)$ *is unique for every state* $s$, *and satisfies*

$$\Delta = \min_s (Q^\star(s, \pi^\star(s)) - \max_{a \neq \pi^\star(s)} Q^\star(s,a)) > 0.$$

Under Assumption 9.1, we derive Assumption 3.2. To that end, suppose $|Q - Q^\star|_\infty \leq \Delta/2$. Then, by definition of the greedy policy, $H^{\pi_Q} = H_{\pi^\star}$, and hence, Assumption 3.2 is trivially satisfied. On the other hand, if $|Q - Q^\star|_\infty > \Delta/2$, then from $|H^{\pi_Q} - H^{\pi^\star}| \leq 2$ it follows

$$|(H^{\pi_Q} - H^{\pi^\star})(Q - Q^\star)|_\infty \leq 2|Q - Q^\star|_\infty \leq \frac{4}{\Delta}|Q - Q^\star|_\infty^2.$$

## 10 ADDITIONAL EXPERIMENTS

In this section, we work with a large discount factor $\gamma = 0.99$, and consider the step-size choices polynomially decaying, LD2Z and PD2Z-$\nu$ with $\nu = 2, 3$. Firstly, we consider $n = 20000$ number of Q-learning iterations, and look at a special case of polynomially decaying step-size, *viz.* the linearly decaying step size $\eta_t = 0.25/t$. Based on $B = 500$ Monte Carlo repetitions, we plot empirical estimates of $\mathbb{E}[|\mathbf{Q}_t - \mathbf{Q}^\star|_\infty]$ against $t \in [n]$ for the LD2Z step-size $\eta_t = 0.25(1 - t/n)$ and PD2Z-$\nu$ step-size choices $\eta_t = 0.25(1 - t/n)^\nu$ with $\nu = 2, 3$, and compare it with empirical estimates of $\mathbb{E}[|\bar{\mathbf{Q}}_t - \mathbf{Q}^\star|_\infty]$ for the linearly decaying step-size, where $\bar{\mathbf{Q}}_t = t^{-1}\sum_{s=1}^{t} \mathbf{Q}_t$ denotes the running Polyak-Ruppert average. It can be seen in Figure 5 that as per our intuition and previous

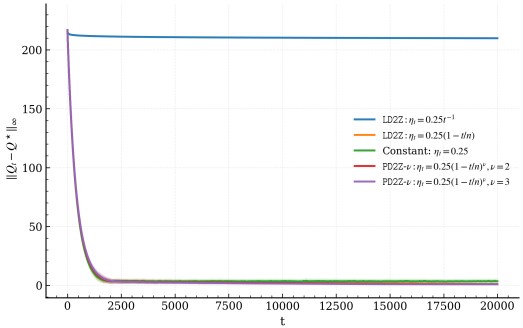

Figure 5: Comparison between different step-size choices.

results, neither the end-term nor the PR-avergaed iterates have converged even after 20000 iterations

for linearly decaying step-sizes; they will eventually converge, and will eventually obtain better asymptotic approximation error compared to LDTZ or PDTZ stepsize choices, but this asymptotic regime kicks in much, much later than is often realistically possible in many scenarios. We can also replicate corresponding versions of Figure 2 for $\gamma = 0.99$ with this particular setting, which we report below.

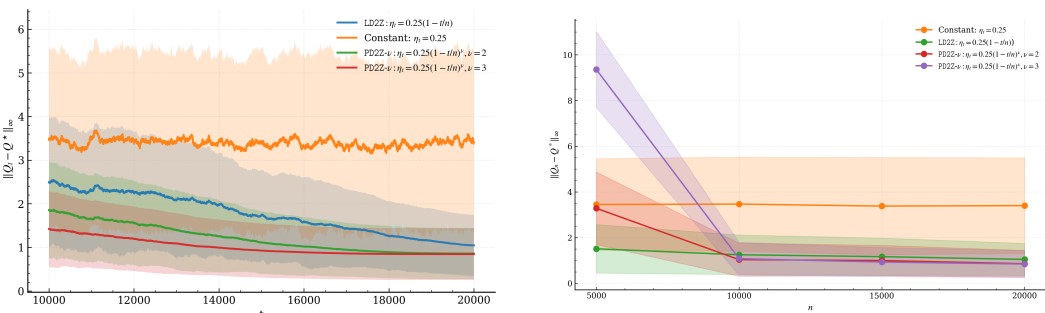

Figure 6: Performance comparison between `LD2Z`, `PD2Z-`$\nu$ with $\nu = 2, 3$ and constant learning schedules.

## 10.1 AFFECT OF LEARNING RATE CONSTANT

To further validate the efficacy of our learning rate schedules, we consider the effect of leading constant $\eta$ in the performance of the Q-iterates. In the following, we consider our $4 \times 4$ gridworld with discount $\gamma = 0.99$, and the following step-sizes: polynomially decaying : $\eta_t = \eta t^{-0.55}$; constant: $\eta_t = \eta$; `LD2Z`: $\eta_t = \eta(1 - t/n)$; `PD2Z-`$\nu$-2: $\eta_t = \eta(1 - t/n)^2$; and `PD2Z-`$\nu$-3: $\eta_t = \eta(1 - t/n)^3$. We vary $\eta \in \{0.1, \ldots, 0.9\}$. For each choice of $\eta$ and learning-rate, we run the Q-learning iterates for $T = 20,000$ episodes, and report the sum of rewards per epsiodes averaged over 100 initial episodes (for the *initial phase*), and 1000 final episodes (for the *asymptotic phase*). The averaged total rewards are further averaged over 500 Monte Carlo runs for stability.

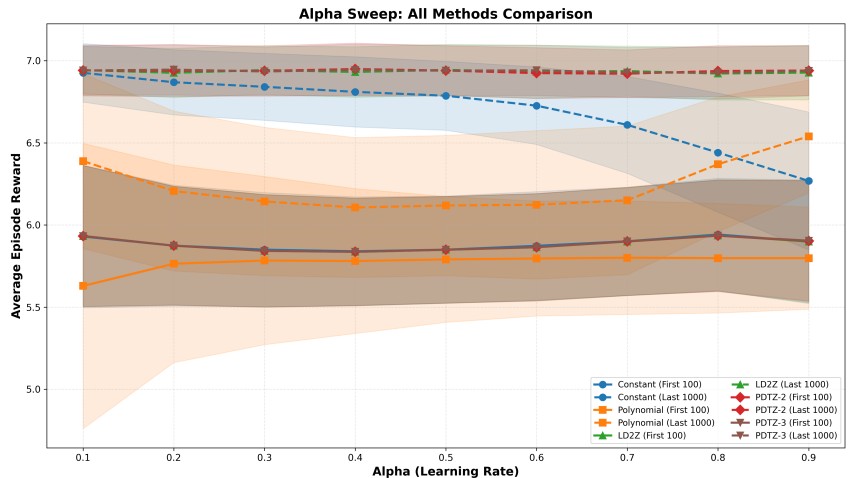

Figure 7: Total sum of rewards on an average reward for the initial phase and asymptotic phase for different learning rates and different $\eta$'s

In Figure 7, the solid lines correspond to the initial phase, and the dashed lines correspond to the asymptotic phase. It is clear that the polynomially decaying step-size is least performing in the initial stage. Moreover, even after 50000 episodes, its asymptotic phase hasn't kicked in. On the other hand, the fact that Q-learning constant learning rate does not converge, is also evident, as larger learning rate constant constant results in increasing bias. In comparison, both the `LD2Z` and `PD2Z-`$\nu$ learning

rates maintain a performance comparable to the constant learning rate in the initial phase, while providing convergence in the asymptotic stage.

## 10.2 ADDITIONAL SIMULATIONS ON CENTRAL LIMIT THEORY

We investigate the asymptotic normality of $\bar{\mathbf{Q}}_n$. For $n = 5000$ and $10,000$, we compute $\mathbf{Q}_{n,n} - \mathbf{Q}^\star$, and project them along 6 randomly chosen directions $u \in \mathbb{S}^{d-1}$. For each random direction $u$, the empirical quantiles of $n^{1/4} u^\top (\bar{\mathbf{Q}}_n - \mathbf{Q}^\star)$ - generated based on $B = 1000$ Monte-Carlo repetitions - are visualized in a QQ-plot against the corresponding quantiles from a standard normal distribution. The asymptotic normality is apparent from the QQ-plot being on a straight line. The accuracy of the scaling $n^{1/4}$ is also evident from the two QQ -plots, corresponding to $n = 5000$ and $n = 10,000$, being virtually identical.

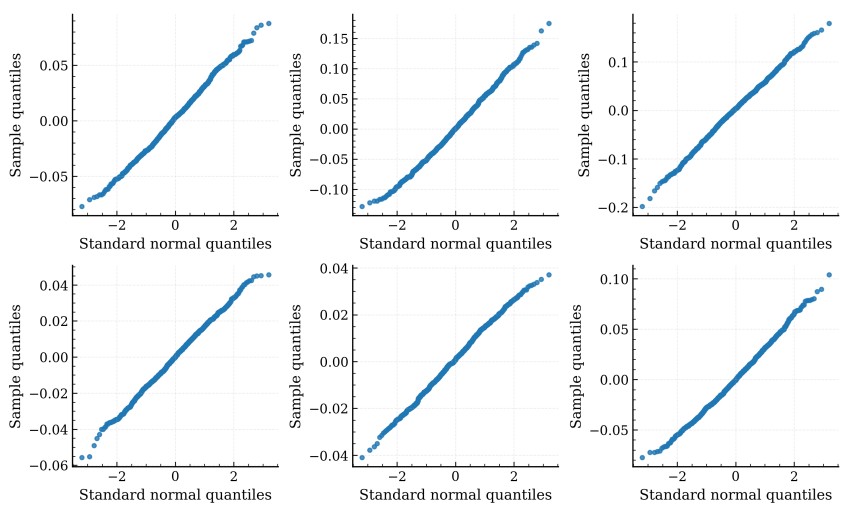

Figure 8: QQ-plots of $n^{1/4} u^\top (\bar{\mathbf{Q}}_n - \mathbf{Q}^\star)$ for randomly generated unit vectors $u$ and $n = 5000$.

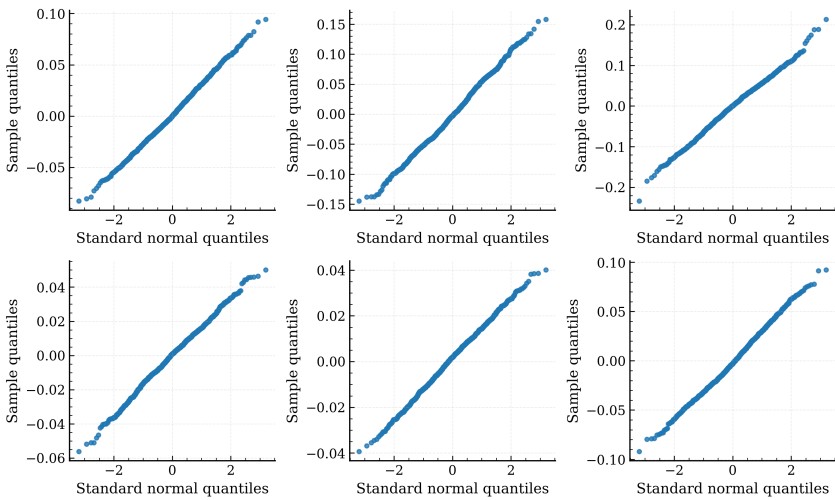

Figure 9: QQ-plots of $n^{1/4} u^\top (\bar{\mathbf{Q}}_n - \mathbf{Q}^\star)$ for randomly generated unit vectors $u$ and $n = 10000$.

## 11 AUXILIARY RESULTS

In this section, we collect some key mathematical arguments that we have repeatedly used throughout our proofs.

**Lemma 11.1.** *Let* $\mathcal{A}_s^t = \prod_{j=s+1}^t (1 - \eta_{j,n} c)$ *for some small* $c \in (0,1)$, *with* $\eta_{s,n} = \eta(1 - \frac{s}{n})^\nu$, $\eta > 0$, $\eta c < 1$ *and* $\nu \geq 1$. *Then for all* $p \geq 1$, $t \in [n]$, *it holds that*

$$
\sum_{s=1}^t \eta_{s,n}^p \mathcal{A}_s^t \leq \begin{cases} C_1(c,\nu,p)\eta_{t,n}^{p-1}, & t \leq n - \dfrac{2}{(c\eta)^{\frac{1}{\nu+1}}} n^{\frac{\nu}{\nu+1}}, \\ C_2(c,\nu,p)n^{-\frac{\nu}{\nu+1}(p-1)}, & t > n - \dfrac{2}{(c\eta)^{\frac{1}{\nu+1}}} n^{\frac{\nu}{\nu+1}}, \end{cases}
$$

*where* $C_1(c,\nu,p)$ *and* $C_2(c,\nu,p)$ *are defined as in Theorem 3.1.*

*Proof of Lemma 11.1.* Our proof proceeds through a series of steps by first establishing a uniform bound on $\mathcal{A}_s^t$, and then carefully establishing control on $\sum_{s=1}^t \eta_{s,n}^p \mathcal{A}_s^t$ on a case-by-case basis. To that end, let $\mathcal{J}(u) = (1 - u/n)$, $u \in [0,n]$. Observe that $u \mapsto \mathcal{J}(u)^\nu$ is a non-increasing function for any $\nu \geq 1$. Therefore, for any $s < t \in [n]$, it follows

$$
\sum_{j=s+1}^t \eta_{j,n} \geq \eta \int_{s+1}^{t+1} \mathcal{J}(u)^\nu \, \mathrm{d}u = \frac{\eta n}{\nu+1}(\mathcal{J}(s+1)^{\nu+1} - \mathcal{J}(t+1)^{\nu+1}) \geq \eta \mathcal{J}(t+1)^\nu(t-s),
$$
(11.1)

where the final inequality in (11.1) follows from the non-increasing property of $\mathcal{J}$. Consequently, one can use (11.1) to derive that

$$
\mathcal{A}_s^t \leq \exp(-c_3 \sum_{j=s+1}^t \eta_{j,n}) \leq \exp(-c_3 \eta \mathcal{J}(t+1)^\nu(t-s)).
$$
(11.2)

This completes the first step of our argument. Moving on, we use (11.2) to derive sharp upper bounds on $\sum_{s=1}^t \eta_{s,n}^p \mathcal{A}_s^t$. This can be approached as follows.

**Case 1.** $t > n - \dfrac{2}{(c_3\eta)^{\frac{1}{\nu+1}}} n^{\frac{\nu}{\nu+1}}$. In this case, we proceed:

$$
\sum_{s=1}^t \eta_{s,n}^p \mathcal{A}_s^t \leq \eta^p \sum_{s=1}^t \mathcal{J}(s)^{\nu p} \exp(-c_3 \frac{\eta n}{\nu+1}(\mathcal{J}(s+1)^{\nu+1} - \mathcal{J}(t+1)^{\nu+1}))
$$

$$
= \eta^p n^{-\nu p} \sum_{s=1}^t (n-s)^{\nu p} \exp\Big(-c_3\eta\frac{n^{-\nu}}{\nu+1}((n-s-1)^{\nu+1} - (n-t-1)^{\nu+1})\Big)
$$

$$
= \eta^p n^{-\nu p} \sum_{k=n-t}^{n-1} k^{\nu p} \exp\Big(-c_3\eta\frac{n^{-\nu}}{\nu+1}((k-1)^{\nu+1} - (n-t-1)^{\nu+1})\Big)
$$

$$
\leq \eta^p n^{-\nu p} \int_{n-t-1}^\infty (u+1)^{\nu p} \exp\Big(-c_3\eta\frac{n^{-\nu}}{\nu+1}(u^{\nu+1} - (n-t-1)^{\nu+1})\Big) \, \mathrm{d}u
$$

$$
\leq \eta^p 4^{\nu p} n^{-\nu p} \exp(\frac{c_3\eta}{\nu+1}\frac{(n-t-1)^{\nu+1}}{n^\nu}) \int_0^\infty (u^{\nu p} + 1) \exp(-c_3\eta\frac{n^{-\nu}}{\nu+1}u^{\nu+1}) \, \mathrm{d}u
$$

$$
\leq 2\eta^p 4^{\nu p} n^{-\nu p} \exp(\frac{2^{\nu+1}}{\nu+1})(\nu+1)^{(p-1)\frac{\nu}{\nu+1}}(c_3\eta)^{-\frac{\nu p+1}{\nu+1}}\Gamma(\frac{\nu p+1}{\nu+1})n^{\frac{\nu}{\nu+1}(\nu p+1)} \quad (11.3)
$$

$$
\leq 2\eta^p 4^{\nu p} \exp(\frac{2^{\nu+1}}{\nu+1})(\nu+1)^{(p-1)\frac{\nu}{\nu+1}}(c_3\eta)^{-\frac{\nu p+1}{\nu+1}}\Gamma(\frac{\nu p+1}{\nu+1})n^{-\frac{\nu}{\nu+1}(p-1)}, \quad (11.4)
$$

where in (11.3) we have invoked $n - t < \dfrac{2}{(c_3\eta)^{\frac{1}{\nu+1}}} n^{\frac{\nu}{\nu+1}}$.

**Case 2:** $t \leq n - \dfrac{2}{(c_3\eta)^{\frac{1}{\nu+1}}} n^{\frac{\nu}{\nu+1}}$.

First observe that,

$$\sum_{s=1}^{t} \eta_{s,n}^{p} \mathcal{A}_s^t \leq \eta^p \sum_{s=1}^{t} \mathcal{J}(s)^{\nu p} \exp(-c_3 \eta \mathcal{J}(t+1)^\nu (t-s))$$

$$\leq \eta^p \sum_{k=0}^{t-s} \mathcal{J}(t-k)^{\nu p} \exp(-c_3 \eta \mathcal{J}(t+1)^\nu k)$$

$$\leq \eta^p \sum_{k=0}^{\infty} \left(\mathcal{J}(t) + \frac{k}{n}\right)^{\nu p} \exp(-c_3 \eta \mathcal{J}(t+1)^\nu k) \tag{11.5}$$

$$\leq \eta^p \frac{c_3 \eta \mathcal{J}(t+1)^\nu}{1 - \exp(-c_3 \eta \mathcal{J}(t+1)^\nu)} \int_0^\infty \left(\mathcal{J}(t) + \frac{u}{n}\right)^{\nu p} \exp(-c_3 \eta \mathcal{J}(t+1)^\nu u) \, \mathrm{d}u \tag{11.6}$$

$$\leq \eta^p \frac{2^{\nu p - 1}}{1 - \exp(-c_3 \eta \mathcal{J}(t+1)^\nu)} \left(\mathcal{J}(t)^{\nu p} + \int_0^\infty \frac{v^{\nu p}}{(c_3 \eta n \mathcal{J}(t+1)^\nu)^p} \exp(-v) \, \mathrm{d}v\right) \tag{11.7}$$

$$\leq \eta^p \frac{2^{\nu p - 1}}{1 - \exp(-c_3 \eta \mathcal{J}(t+1)^\nu)} \left(\mathcal{J}(t)^{\nu p} + (c_3 \eta n \mathcal{J}(t+1)^\nu)^{-p} \Gamma(\nu p + 1)\right), \tag{11.8}$$

where (11.5) follows from noting $\mathcal{J}(t-k) = \mathcal{J}(t) + \frac{k}{n}$; (11.6) derives from an application of Lemma 11.2; (11.7) is obtained by the elementary inequality $(x+y)^q \leq 2^{q-1}(x^q + y^q)$ for $q \geq 1$. Finally, in (11.8), $\Gamma(\cdot)$ denotes the Gamma function. The two terms in (11.8) following the leading constants are particularly interesting; the first term increases with $t$, and the second term decays with $t$. The interplay between these two terms will naturally lead to two regions on which the rates will be controlled case-by-case.

Now, recall that in this particular regime, it is immediate that $c_3 \eta n \mathcal{J}(t)^\nu \geq \frac{2^{\nu+1}}{\mathcal{J}(t)}$. Moreover, since $n$ is sufficiently large such that $\frac{2}{(c_3 \eta)^{\frac{1}{\nu+1}}} n^{\frac{\nu}{nu+1}} > 2$, it follows that in this regime, $\mathcal{J}(t+1) \geq \mathcal{J}(t)/2$. Therefore,

$$\mathcal{J}(t)^{\nu p} + (c_3 \eta n \mathcal{J}(t+1)^\nu)^{-p} \Gamma(\nu p + 1) \leq \mathcal{J}(t)^{\nu p}(1 + 2^{-p}\Gamma(\nu p + 1)),$$

which, when plugged in (11.8), implies that

$$\sum_{s=1}^{t} \eta_{s,n}^{p} \mathcal{A}_s^t \leq \eta^p \frac{2^{\nu p - 1}}{1 - \exp(-c_3 \eta \mathcal{J}(t+1)^\nu)} \mathcal{J}(t)^{\nu p}(1 + 2^{-p}\Gamma(\nu p + 1))$$

$$\leq \frac{2^{\nu(p+1)}(1 + 2^{-p}\Gamma(\nu p + 1))}{c_3} \eta_{t,n}^{p-1}, \tag{11.9}$$

where in the final inequality we have used $c_3 \eta < 1$ to deduce

$$1 - \exp(-c_3 \eta \mathcal{J}(t+1)^\nu) \geq \frac{c_3 \eta \mathcal{J}(t+1)^\nu}{2} \geq \frac{c_3 \eta \mathcal{J}(t)^\nu}{2^\nu}.$$

Finally, (11.4) and (11.9) completes the proof. $\qquad\square$

**Lemma 11.2.** *Let $f : \mathbb{R} \to \mathbb{R}_+$ be a non-decreasing function and let $\kappa > 0$ be a constant such that $\sum_{n=0}^{\infty} f(n) \exp(-\kappa n) < \infty$. Then*

$$\sum_{n=0}^{\infty} f(n) \exp(-\kappa n) \leq \frac{\kappa}{1 - \exp(-\kappa)} \int_0^\infty f(u) \exp(-\kappa u) \, du.$$

*Proof.* Since $f$ is non-decreasing, hence for every $n \in \mathbb{N}$,

$$f(n) \exp(-\kappa n) = \frac{\kappa}{1 - \exp(-\kappa)} f(n) \int_n^{n+1} \exp(-\kappa u) \, \mathrm{d}u \leq \frac{\kappa}{1 - \exp(-\kappa)} \int_0^\infty f(u) \exp(-\kappa u) \, \mathrm{d}u.$$

$$\square$$