# OpenReview forum: "Sharp asymptotic theory for Q-learning with \texttt{LD2Z} learning rate and its generalization"
_ICLR.cc/2026/Conference — ICLR 2026 Poster_

### Official Review · Reviewer_vcVR · 2025-10-27

**Soundness:** 3
**Presentation:** 3
**Contribution:** 2
**Rating:** 4
**Confidence:** 4

**Summary:**

This paper develops a sharp asymptotic theory for Q-learning with the LZ2D step size schedule.

The authors establish several asymptotic results, including almost sure convergence, functional central limit theorems, and last-iterate convergence rates.

The paper highlights that the proposed LZ2D step size achieves improved convergence behavior compared to classical polynomial or constant step-size schemes.

Empirical results demonstrate the performance of LZ2D on synthetic Q-learning experiments, supporting its claimed theoretical benefits.

**Strengths:**

1. **Comprehensive theoretical contributions.**

    The paper provides a complete asymptotic characterization — including almost sure convergence, convergence rates for the last iterate, and uniform approximation (Section 4). This set of results is new for Q-learning with the particular step size and strengthens the theoretical foundation of this step size choice.

2. **Clear theoretical presentation.**

    The main theorems are well organized and connected logically. The assumptions are explicitly stated, and the paper maintains consistency with the established stochastic approximation framework, which aids readability despite the technical depth.

3. **Empirical validation.**

    Figures 1 and 3 compare different step-size rules and suggest that the proposed LZ2D schedule achieves faster or more stable convergence. The figures are clearly presented and help visualize the claimed benefits.

**Weaknesses:**

1. **LZ2D makes an online algorithm effectively offline.**

    The LZ2D step-size schedule $\eta_{t,n}$ depends explicitly on the total sample size $n$, making it unsuitable for online learning. This fundamentally alters the nature of Q-learning, which is originally designed for streaming data.
    * If new samples continue arriving after the prescribed $n$, it is unclear how to update $\eta_{t, n}$.
    * The paper should discuss whether asymptotic results remain valid if $n$ grows or is mis-specified.
    * This issue is critical for any claim of practical superiority.

    From this perspective, LZ2D transforms Q-learning from an online algorithm to an offline one, which undermines its applicability in reinforcement learning.

2. **Trade-off between last-iterate convergence and asymptotic normality is missing.**

    The paper claims both fast last-iterate convergence and asymptotic efficiency, but these are known to be mutually exclusive under standard stochastic approximation theory.
    * As shown in ROOT-SGD (Li, Mou, Wainwright, Jordan, 2020), the last iterate typically converges at rate $O(1/T)$, while asymptotic normality applies to the averaged iterate with $\sqrt{T}$ scaling.
    * In this paper, Theorem 3.5 shows that for PD2Z-$\nu$ step sizes, the scaling becomes $n^{\nu/(2(\nu+1))}$, slower than $\sqrt{n}$, except in the limit $\nu \to \infty$. This trade-off is not acknowledged. The current presentation gives the impression that LZ2D dominates all prior step-size rules, which is mathematically misleading.

       **Reference:** Li C J, Mou W, Wainwright M, et al. Root-sgd: Sharp nonasymptotics and asymptotic efficiency in a single algorithm[C]//Conference on Learning Theory. PMLR, 2022: 909-981.

3. **Experimental comparisons are incomplete and possibly biased.**

    * Figure 1 compares LZ2D with polynomial and constant step sizes but does not include linearly decaying step sizes ($\eta_t \propto 1/t$), which have theoretical guarantees for Q-learning (Li et al., Operations Research, 2024).
    * The paper also omits analysis of averaged iterates, which are standard when polynomial decay is used.
    * The hyperparameters (e.g., exponent $\alpha$ in polynomial decay) are not reported. Without tuning details, the apparent dominance of LZ2D may reflect unfair parameter choices rather than intrinsic superiority. Overall, the empirical claims in Figure 1 and 3 appear “too good to be true” without further clarification or ablation.

        **Reference:** Li G, Cai C, Chen Y, et al. Is Q-learning minimax optimal? a tight sample complexity analysis[J]. Operations Research, 2024, 72(1): 222-236.

4. **Unexplained discrepancy in Brownian motion approximation (Figure 3).**

     The right panel of Figure 3 shows a large performance gap between the LZ2D method and the Brownian motion benchmark, despite both satisfying the same functional CLT.
    * The paper does not explain why the LZ2D trajectory aligns so much better.
    * This may be due to slow convergence of remainder terms, but no quantitative discussion is given.
    * Additionally, there is an inconsistency: Line 455 defines the measurement as $\max_{k_n \le t \le n} |\sum_{l=t}^n (Q_l^c - Q^*)|{\infty}$, while Figure 3 uses $\max_{1 \le t \le n}$. Which definition is correct, and does LZ2D introduce the $k_n$ term while the Brownian benchmark does not? Clarifying this is important, since it affects the interpretation of the approximation quality.

**Questions:**

Please see the Weaknesses part. I will increase my point if the weaknesses are addressed well.

---

> ### Author Response · Authors · 2025-11-21
> **Response to Reviewer vcVR**
>
> Firstly, we thank the reviewer for acknowledging our comprehensive theoretical results, as well as the clarity and novelty of our work. We also thank the reviewer for raising these insightful questions, which we answer in the following in separate comments. We also appreciate the reviewers openness to increasing the score upon adequate rebuttal.
>
> > **L2ZD makes an online algorithm effectively offline.**  The L2ZD step-size schedule $\eta_{t,n}$ depends explicitly on the total sample size $n$, making it unsuitable for online learning. This fundamentally alters the nature of Q-learning, which is originally designed for streaming data.
> > - If new samples continue arriving after the prescribed $n$, it is unclear how to update $\eta_{t,n}$.
> > - The paper should discuss whether asymptotic results remain valid if $n$ grows or is mis-specified.
> > - This issue is critical for any claim of practical superiority.
> > From this perspective, L2ZD transforms Q-learning from an online algorithm to an offline one, which undermines its applicability in reinforcement learning.
>
> We agree that we require total sample size $n$ to be known in advance. We acknowledge this as the main limitation of the L2ZD schedule when applied to Q-learning. However, we would like to argue that the algorithm is still online, as the samples are processed one-by-one in a streaming manner. We do not require prior knowledge of the entire dataset.
>
> Since the current L2ZD schedule cannot accommodate new samples that arrive after the total sample size, we focus on the case where $n$ is mis-specified. Let $n_{0} \leq n$ denote the true sample size, while $n$ is used in the L2ZD step-size schedule. Then, as long as the mis-specification satisfies
> $n - n_{0} \leq \alpha \sqrt{n}$
> for some constant $\alpha \in (0, 1)$, our asymptotic results remain valid. We have included this comment in the revised manuscript.
>
> Your comments, however, motivate us to propose a piecewise version of the L2ZD step-size schedule that can handle online streaming data. Specifically, for a relatively large integer $T$, we define $\eta_{0} = \eta$ and update the step size by
>
> $$
> \eta_{t} = \eta c_i \left(1- \frac{t-i^{\alpha}}{(i+1)^{\alpha} - i^{\alpha}}\right),
> \quad i^\alpha \leq t < (i+1)^\alpha, \quad i = 1, 2, \ldots
> $$
>
> Here, $(\{c_i\})_{i \geq 1}$ is some sequence decaying to zero, and $\alpha$ is appropriately chosen. For each $i \geq 1$, the step size linearly decays from $i^\alpha$ to $(i+1)^\alpha$. Such block-based methods are prevalent in the context of covariance estimation [1], bootstrap [2], [3] for SGD as well as for processes with inherent non-stationarity.
>
> Based on familiarity with this literature, we believe the probabilistic tools developed in the current paper can offer insights into the theoretical analysis of this piecewise schedule; however, a full study of the method would be quite involved and require substantial additional work. In particular, a new version of the *tail Polyak–Ruppert averaging estimator* would need to be designed, and a new Gaussian approximation result would have to be established.
>
> Given the scope of these extensions, we plan to leave a systematic investigation of the proposed piecewise L2ZD step-size schedule for future work. We appreciate your thoughtful comments, which have inspired new ideas and potential improvements beyond the current paper.
>
> **References:**
> [1] Zhu, Wanrong, Xi Chen, and Wei Biao Wu. *Online covariance matrix estimation in stochastic gradient descent.* JASA 118.541 (2023): 393–404.
> [2] Dette, Holger, and Carina Graw. *Uncertainty quantification by block bootstrap for differentially private stochastic gradient descent.* arXiv:2405.12553 (2024).
> [3] Bühlmann, Peter. *Bootstraps for time series.* Statistical Science (2002): 52–72.

---

> > ### Author Response · Authors · 2025-11-21
> > **Ctd.**
> >
> > > **Trade-off between last-iterate convergence and asymptotic normality is missing.**
> >   The paper claims both fast last-iterate convergence and asymptotic efficiency, but these are known to be mutually exclusive under standard stochastic approximation theory.
> >  > - As shown in ROOT-SGD (Li, Mou, Wainwright, Jordan, 2020), the last iterate typically converges at rate $O(1/T)$, while asymptotic normality applies to the averaged iterate with $\sqrt{T}$ scaling.
> >   > - In this paper, Theorem 3.5 shows that for PD2Z-$\nu$ step sizes, the scaling becomes $n^{\nu/(2(\nu+1))}$, slower than $\sqrt{n}$, except in the limit $\nu \to \infty$. This trade-off is not acknowledged. The current presentation gives the impression that L2ZD dominates all prior step-size rules, which is mathematically misleading.
> >
> >   > **Reference:** Li C J, Mou W, Wainwright M, et al. Root-sgd: Sharp nonasymptotics and asymptotic efficiency in a single algorithm[C]//Conference on Learning Theory. PMLR, 2022: 909-981.
> >
> >
> >   Thank you for raising this very pertinent point. The ROOT-SGD can be thought of as a version momentum-assisted SGD, explaining its enhanced practical performance, but its last iterate convergence of $1/T$ is not unique; it can also be achieved by linearly decaying step-sizes $\eta_t \propto 1/t$. In particular, it is known for both SGD and Q-learning that with polynomially decaying step-sizes $\eta_t \propto t^{-\alpha}$, the last iterate converges as $1/T^{\alpha}$ in MSE, and the PR-averaged version has a scaling $\sqrt{T}$.
> >
> >   Therefore, you are indeed right- our Theorem 3.5 indeed reflects a trade-off in the sense that its asymptotic scaling is $n^{\nu/(2(\nu+1))}$ can never be $\sqrt{n}$ when $\nu$ is finite. However, the point of our theory in this article is two-fold: (i) even though our scheduler is not efficient when we look at PR-averaged versions, our rate, $n^{-\nu/(2(\nu+1))}$ is comparable to the final iterate rate $n^{-\alpha/2}$. In this sense, there is no sub-optimality in the asymptotic regime. (ii) Even in the Polyak-ruppert averaged case, notwithstanding the sub-optimality of a polynomial order that it obtains in the asymptotic regime, the speed-up is immense (in the exponential order) in the initial phase: ($\exp(-T)$ versus $(\exp(-T^{1-\alpha})$). This is also explained in Remark 3.1. That is why we termed this scheduler as inheriting a “best of both worlds’’ property. We do not claim that it mathematically dominates all prior step-size rules, and we are sorry for the confusion caused. The polynomially decaying step-sizes retains its asymtotic efficiency, but at least in the case of Q-learning, the effect of its suboptimality in the initial phase is immense; in almost all simulation exercises, the asymptotic regime for these step-sizes kicks in only for super-large values of $n$. On the other hand, the constant step-sizes, even though quite fast to get to the vicinity of the optimal $Q$-function, do not actually converge there. Our step-size, even though sacrificing the asymptotic efficiency of a PR-averaged version, inherits both a fast initial decay, and an asymptotic convergence polynomial in $T$. At your suggestion, we are prepared to clarify this in the main text.

---

> ### Author Response · Authors · 2025-11-21
> **Ctd.**
>
> > **Experimental comparisons are incomplete and possibly biased.**
> > - Figure 1 compares L2ZD with polynomial and constant step sizes but does not include linearly decaying step sizes ($\eta_t \propto 1/t$), which have theoretical guarantees for Q-learning (Li et al., Operations Research, 2024).
>  > - The paper also omits analysis of averaged iterates, which are standard when polynomial decay is used.
>   > - The hyperparameters (e.g., exponent $\alpha$ in polynomial decay) are not reported. Without tuning details, the apparent dominance of L2ZD may reflect unfair parameter choices rather than intrinsic superiority. Overall, the empirical claims in Figure 1 and 3 appear “too good to be true” without further clarification or ablation.
>
>   > **Reference:** Li C, Gai C, Chen Y, et al. Is Q-learning minimax optimal? a tight sample complexity analysis. *Operations Research*, 2024, 72(1): 222–236.
>
>   Thank you for pointing out the omitted details in Figure 1, and giving us the opportunity assure you that our reported figures are not results of unfair parameter choices. In particular, just like in Figures 2 and 3, our results reported results for polynomial decay step-sizes are with learning rate $0.05 t^{-0.65}$. We have also run experiments with learning rates $\eta t^{-0.55}$ where $\eta \in \{0.1, 0.2, \ldots, 0.9\}$. We omitted comparing with polynomially decaying step-sizes in our numerical experiments sections precisely because we observed, that even with $50,000$ episodes (for asynchronized Q-learning) or $50,000$ iterates (for synchronized Q-learning), the Q-iterates with Polynomial step-sizes haven't converged (i.e. the asymptotic regime hasn't yet kicked in). Therefore, we shouldn't expect anything different if we do the PR-averaged version or the linearly decaying step-size $\eta_t \approx 1/t$. Despite the presence of theoretical guarantees, similar observations were reported in Zhang & Xie [1], where they write:
> >Particularly for the more complicated the MDP, as shown in Figure 1(b), iterates with diminishing stepsize converge slowly.
>
> Note that their Figure 1.(b) corresponds to the same $4\times 4$ gridworld setting as ours, with the discount factor $\gamma=0.99$ for the Q-learning algorithm. Nevertheless, in cognizance of your comments, we ran the Q-learning iterates for $\eta_t \propto 1/t$, and analyzed the error of its PR-averaged version. We report the figures in Section 9, Figures 5–6, of the revised manuscript. As per our intuition and previous results, neither the end-term nor the PR-avergaed iterates have converged even after $20000$ iterations for linearly decaying step-sizes; they will eventually converge, and will eventually obtain better asymptotic approximation error compared to LDTZ or PDTZ stepsize choices, but this asymptotic regime kicks in much, much later than is often realistically possible in many scenarios. In **Appendix Section 9**, we have included more simulation studies at the insistence of other reviewers, and have clarified the details therein.
>
> **References**: Yixuan Zhang and Qiaomin Xie. Constant stepsize q-learning: Distributional convergence, bias and
> extrapolation. arXiv preprint arXiv:2401.13884,

---

> > ### Author Response · Authors · 2025-11-21
> > **Ctd**
> >
> > >**Unexplained discrepancy in Brownian motion approximation (Figure 3).**
> >   The right panel of Figure 3 shows a large performance gap between the L2ZD method and the Brownian motion benchmark, despite both satisfying the same functional CLT.
> >  > - The paper does not explain why the L2ZD trajectory aligns so much better.
> >  > - This may be due to slow convergence of remainder terms, but no quantitative discussion is given.
> >  > - Additionally, there is an inconsistency: Line 455 defines the measurement as $\max_{k \le t_n} \|\sum_{i=1}^k (Q_i - Q^\star) \|\_\infty$, while Figure 3 uses $\max_{1 \le k \le t_n}$. Which definition is correct, and does L2ZD introduce the $k_n$ term while the Brownian benchmark does not? Clarifying this is important, since it affects the interpretation of the approximation quality.
> >
> > We are again extremely sorry for the confusion caused, but the Figure 3 does not deal with the performance gap between L2ZD and the Brownian motion benchmark. We attempt to briefly explain the purpose of this numerical experiment in the following.
> >
> > In Theorems 4.1 and 4.2, we provide a particular form of strong invariance principle for Q-iterates with both polynomial and L2ZD step-sizes. For each of these step-size choices, another strong invariance principle in the form of Brownian motion approximation can also be proved; for example, see Theorem 3.1 in Li et. al. [1]. The purpose of Figure 3 is to compare the approximation accuracy of our particular invariance principle (which has a particular form of asymptotic Gaussian process) and the invariance principle based on Brownian motion benchmark.
> >
> > In particular, in Figure 3.(a), we look at our invariance principle in Theorem 4.1 for the L2ZD step-size, and compare the quantiles based on our Gaussian approximations to the actual theoretical quantiles. Since there is no Brownian motion-based approximation known for this step-size choice, we do not compare with this here. The alignment in this figure is not of the L2ZD-based trajectory with the Brownian motion, but of the Gaussian approximation of the L2ZD-based trajectory to the original L2ZD-based trajectory.
> >
> > On the other hand, in Figure 3.(b), we consider our invariance principle in Theorem 4.2 for Q-iterates with polynomially decaying step-size. In this case, apart from our invariance principle, a Brownian-motion based approximation is also known (Theorem 3.1 in Li et. al. [1]). We would like to clarify that the $1\leq l \leq n$ here is not an inconsistency, since the invariance principle for polynomially decaying step-sizes hold for the entire range $1\leq l \leq n$ (in contrast to the case for L2ZD step-size choice, which holds for $\max_{k_n \leq l \leq n}$. Moreover, Figure 3.(b) shows that the quantiles based on our Gaussian approximation is much more accurate than a Brownian-motion based approximation. The key reason behind this is indeed sharper remainder terms, in a property which is dubbed “the covariance matching property’’ (Bonnerjee et. al. [2]). Its intuition is provided in the discussion immediately succeeding Theorem 4.2, hugging the equations (4.3) and (4.4).
> >
> > We hope that this clears the confusion up. We have also briefly clarified this in Section 5.3 of the revised manuscript.
> >
> > **References:**
> > [1] Xiang Li, Wenhao Yang, Jiadong Liang, Zhihua Zhang, and Michael I Jordan. “A statistical analysis of polyak-ruppert averaged q-learning”. AISTATS 2023.
> > [2] Bonnerjee, Soham, Sayar Karmakar, and Wei Biao Wu. “Gaussian approximation for nonstationary time series with optimal rate and explicit construction.” The Annals of Statistics 52.5 (2024): 2293–2317.

---

> ### Comment · Reviewer_vcVR · 2025-11-27
>
> Thanks for the authors’ detailed feedback. Most of my concerns have been resolved.
>
> However, I’m still uncertain whether I fully understand why the Brownian motion approximation performs worse than the proposed Gaussian approximation. I think the reason is that the author didn't formulate their bootstrap-based algorithm for critical value estimation clearly or explicitly in the paper.
>
> I have to guess the reason. My current understanding—and speculation—is as follows:
>
> - Li et al. (2023b) computes critical values using the partial sums of a Brownian motion process.
>
> - In contrast, this paper proposes to use a simulated iterative process (defined in (4.1)) and then compute critical values based on that process instead.
>
> The key difference is that the proposed method preserves a finer dependency structure between consecutive iterates, while this structure is effectively ignored in Li et al. (2023b). As a result, when the sample size is limited, the proposed simulation should offer a more accurate representation of the dependence pattern in the data, which explains its improved performance in Figure 3 (right).
>
> Is my current understanding correct? On the other hand, could I say, if the sample size increases a lot and the initial error decay faster enough, then the alignment of the Brownian motion approximation would be improved?
>
> By the way, one minor point: In Figure 7, the legend labels for the initial phase and the asymptotic phase appear to be identical.

---

> ### Author Response · Authors · 2025-11-29
> **Reply to Reviewer vcVR**
>
> We thank the reviewer for the kind response, and the acknowledgment of our detailed feedback. We also appreciate the opportunity to provide further clarification regarding the proposed Gaussian approximations.
>
> First we provide a “wrapper’’ bootstrap algorithm to further motivate our Gaussian approximation results. Suppose you are interested in inference for $f(Q_n)$.
>
> - Simulate $B$ many copies
>
> $$
> ( \tilde{Y}\_t^{1} )_{t=1}^n, \ldots, ( \tilde{Y}\_t^{B} )\_{t=1}^n,
> $$
>
>  where $B$ is the number of bootstrap samples, and for each $b \in [B]$, $(\tilde{Y}\_t^{\,b})_{t=1}^n \sim (\tilde{Y}_t)$ from Theorem 4.2.
>
> - Let the cut-off be $\hat{q}\_n = Q_{1-\alpha}(\{ f(Y_n^1), \ldots, f(Y_n^B) \})$.
>
> - Based on $\hat{q}_n$, perform statistical tests / build confidence intervals on $f(Q_n)$.
>
> As mentioned in the discussion following Theorem 4.2, we call our Gaussian approximation as ``covariance matching approximation''. This is because, as you have correctly surmised, that our Gaussian approximation results maintain a finer dependency structure than the Brownian motion-based approximation, which has independent increments and do not respect the non-stationarity inherent in the Q-learning trajectory. To explain it further, we take a look at [1], where the authors introduce similar covariance-matching approximations for some particular class of non-stationary time-series. In their case, they also had access to a Brownian motion approximation with the exact optimal rate as that of the covariance-matching approximation (see Theorems 2.2 and 2.4 therein). Even in their case, they observe better finite sample performance from covariance-matching approximation, and they conjecture that it is due to the preservation of the exact dependency structure of the covariance-matching approximations. Therefore, the reviewer is absolutely correct in the understanding.
>
> In our case, we note another finer point while comparing the Brownian motion and covariance matching based approximations. Our approximations achieve a rate of $o(n^{1/p})$ with $p$ moments, which is known to be optimal in time-series literature. On the other hand, only a $o(\sqrt{n})$ rate is known for the Brownian-motion based approximation, which may not be sharp [2]. It is conceivable that due to the Brownian motion based approximation not maintaining the dependency structure, not all of the leading terms are not accounted for, and therefore, the Brownian motion based approximation may not even achieve the optimal rate, explaining the better performance further. However, we emphasize that even if the Brownian motion approximation can be proved to also achieve the optimal rate, its asymptotic nature means it will match our covariance-matching approximations' performance **only when the sample size is too large**. In this regard, our covariance matching approximation can be understood to be maintaining the optimality while preserving the dependency structure, presenting an altogether better alternative.
>
> > By the way, one minor point: In Figure 7, the legend labels for the initial phase and the asymptotic phase appear to be identical.
>
>  Thank you for pointing this out. This was Python’s default labeling, even though we clarified in writing that the solid lines correspond to the initial phase, and the dashed lines correspond to the asymptotic phase. We will make sure this is corrected in the final version.
>
> [1] Bonnerjee, Soham, Sayar Karmakar, and Wei Biao Wu. “Gaussian approximation for nonstationary time series with optimal rate and explicit construction.’’ Annals of Statistics 52.5 (2024): 2293–2317.
>
> [2] Xiang Li, Wenhao Yang, Jiadong Liang, Zhihua Zhang, and Michael I. Jordan. “A statistical analysis of Polyak–Ruppert averaged Q-learning.’’ AISTATS 2023.

---

### Official Review · Reviewer_WVUn · 2025-10-29

**Soundness:** 4
**Presentation:** 3
**Contribution:** 3
**Rating:** 8
**Confidence:** 4

**Summary:**

This paper studied the theoretical properties of tabular synchronous Q-learning with linear decay to zero (LD2Z) and more general power-law decay to zero (PD2Z) step sizes.
They first provided non-asymptotic L^p error bound, which showcases that, comparing to polynomial decaying step sizes, the error bound of PD2Z could rapidly decay from initialization (an advantage of constant step sizes). Numerical experiments were provided to support the theoretical results. The paper also showed the asymptotic normality and the stronger time-uniform Gaussian approximation of the tail Polyak-Ruppert averaging of their algorithms.

**Strengths:**

The paper is well-written and clearly presents the significance of the research and its contributions.
Specifically, the step size of LD2Z (or more general PD2Z) is more practical in real-world applications, such as in generative models like LLMs. However, theoretical analysis of Q-learning algorithms with this type of step size has remained underdeveloped, and this paper uses rigorous theoretical results to demonstrate that the LD2Z step size strategy can simultaneously retain the advantages of constant step sizes (fast initial convergence) and polynomial decay step sizes (asymptotic normality of the tail average).
Additionally, the paper provides a strong approximation result that is stronger than the functional CLT-like results reported in previous works.
Extensive numerical experiments further validate the theories proposed in this study.
Overall, this represents a very solid theoretical work.

**Weaknesses:**

1. Goldreich et al. (2025) have provided the L2 non-asymptotic error bounds and asymptotic properties of the LD2Z step size for Stochastic Gradient Descent (SGD) under the strong convexity condition. It is recommended that the authors clarify the differences and innovations in the proof techniques of this paper compared to previous works.

2. To my understanding, although the tail-averaged Q-learning with LD2Z step size exhibits asymptotic normality, its asymptotic variance is presumably not asymptotically optimal—in contrast, the asymptotic variance under polynomial decay step sizes is optimal. It would be beneficial if the authors could illustrate, either through theoretical analysis or experimental results, the discrepancy in asymptotic variance between the two step size strategies.

**Questions:**

see weakness

---

> ### Author Response · Authors · 2025-11-21
> **Response to Reviewer WVUn**
>
> We thank the reviewer for the encouraging words on our novelty, theoretical rigor, and extensive numerical experiments, as well as the strong recommendation. In the following, we address the weaknesses one-by-one.
>
> > **Goldreich et al. (2025) have provided the L2 non-asymptotic error bounds and asymptotic properties of the LD2Z step size for Stochastic Gradient Descent (SGD) under the strong convexity condition. It is recommended that the authors clarify the differences and innovations in the proof techniques of this paper compared to previous works.**
>
> The result of Goldreich et. al. (2025) is a $\mathcal{L}_2$ error bound, which albeit novel, does not facilitate statistical inference. In that aspect, our central limit theory and strong invariance principle present a principled direction to statistical inference. The mathematical techniques to derive these results are quite different, as one usually requires careful approximations via martingale difference sequences to infer such optimal Gaussian approximations. Moreover, as we mention in the response to the reviewer 632X, with moderate amount of effort, one can strive to produce similar results for SGD- which in itself is an interesting direction- but there are some clear differences when it comes to Q-learning. For example, in Q-learning, there is no convex potential whose gradient coincides with Bellman residuals, which prevents construction of appropriate Lyapunov functionals.
>
> > **To my understanding, although the tail-averaged Q-learning with LD2Z step size exhibits asymptotic normality, its asymptotic variance is presumably not asymptotically optimal—in contrast, the asymptotic variance under polynomial decay step sizes is optimal. It would be beneficial if the authors could illustrate, either through theoretical analysis or experimental results, the discrepancy in asymptotic variance between the two step size strategies.**
>
>  Even though the asymptotic variance under polynomial decay step-size has been well known to be optimal, the main point we want to convey is that the asymptotic regime kicks in much later, for only very large $n$, which renders practical simulations out-of-reach. For example, with our $4\times 4$ gridworld setting, to achieve the asymptotic regime even with a small discount $\gamma=0.1$, and polynomial step-size $\eta_t=0.25 t^{-0.52}$, we require about $n=100,000$ iterations. In the following, we report the standard deviation, based on $50$ Monte Carlo iterations, of $\sqrt{n}|\bar{\boldsymbol{Q}}_n - \boldsymbol{Q}^\star|\_\infty$ for Polynomial decay, and $n^{1/4}|\tilde{\boldsymbol{Q}}_n - \boldsymbol{Q}^\star|\_\infty$ for LDTZ step-size, where $\tilde{\boldsymbol{Q}}_n$ denotes the tail PR-average. Note that the scaling factors are different. Clearly, the iterates with polynomially decaying step-sizes are much more stable in terms of asymptotic variance, but we would like to remind the reviewer that the point of using LDTZ, despite its apparent asymptotic sub-optimality, is its improved performance as long as $n$ is not super, super large.
>
> | Polynomial-decay                        | LDTZ                          |
> |------------------------------------------|-------------------------------|
> | mean = 0.0144467, std = 0.000704         |    mean = 0.012488, std = 0.089784 |

---

> > ### Comment · Reviewer_WVUn · 2025-11-27
> >
> > Thanks for your response, i'll keep my current score unchanged.

---

### Official Review · Reviewer_HuHH · 2025-10-31

**Soundness:** 3
**Presentation:** 3
**Contribution:** 3
**Rating:** 6
**Confidence:** 2

**Summary:**

The paper develops sharp theory for Q‑learning under a power‑law decay‑to‑zero stepsize schedule. This work then establishes explicit non‑asymptotic error bounds, a central‑limit theorem for a Polyak–Ruppert averaging estimator, and a time‑uniform strong invariance principle. Experiments on a 4×4 FrozenLake‑style gridworld corroborate the theory. This work provides the first non‑asymptotic analysis of LD2Z for Q‑learning.

**Strengths:**

The main strength of this work lies in providing the first non asymptomatic analysis of Q-learning with PD2Z step-size schedules. The analysis is clear, sound, and explains why LD2Z achieves both fast early convergence and asymptotic stability.

**Weaknesses:**

The main weakness of this work lies in its empirical evaluation, while this work is mainly concerned with learning theory, tabular Q-learning is simply enough to run on any modern laptop (or colab). Ideally, using the publicly available code to generate figure 6.9 from the book by Sutton and Barto 2018 with your learning rate scheduler would be beneficial.

**Questions:**

comments

1. line 054: li et al is missing a year.
2. line 241: aforementioned, decay followed by --> decay is followed by

---

> ### Author Response · Authors · 2025-11-15
> **A clarification**
>
> We thank the reviewer for the detailed comments and the encouraging words. Before we provide our official response, we would like to request a clarification. It seems that there is no Figure 6.9 in this book http://incompleteideas.net/book/RLbook2020.pdf. We would very much appreciate if the reviewer can explain more about the type of figures mentioned in the reviews, and we will strive to provide appropriate experimental results. We also thank the reviewer for the detailed perusal to weed out the typos; we have further proofread our article to correct all such errors.

---

> > ### Comment · Reviewer_HuHH · 2025-11-15
> >
> > Ahh my bad, I mean figure 6.3 (below equation 6.9). Thanks for pointing this out.

---

> ### Author Response · Authors · 2025-11-21
> **Response to Reviewer HuHH**
>
> We thank you again for raising this question. We warmly welcome any further suggestions you may have to improve the paper. Finally, in case you find this rebuttal adequate, we humbly request you to consider increasing your score.
>
> > **The main weakness of this work lies in its empirical evaluation, while this work is mainly concerned with learning theory, tabular Q-learning is simply enough to run on any modern laptop (or colab). Ideally, using the publicly available code to generate figure 6.9 from the book by Sutton and Barto 2018 with your learning rate scheduler would be beneficial.**
>
>  In the Section 9.1 of the revised manuscript, we consider our $4\times 4$ gridworld with discount $\gamma=0.99$, and the following step-sizes: polynomially decaying : $\eta_t=\eta t^{-0.55}$; constant: $\eta_t=\eta$; `LDTZ`: $\eta_t= \eta(1-t/n)$; `PDTZ`-2: $\eta_t= \eta(1-t/n)^2$; and `PDTZ`-3: $\eta_t= \eta(1-t/n)^3$. We vary $\eta \in \{0.1,\ldots, 0.9\}$. For each choice of $\eta$ and learning-rate, we run the Q-learning iterates for $T=20,000$ episodes, and report the sum of rewards per epsiodes averaged over $100$ initial episodes (for the *initial phase*), and $1000$ final episodes (for the *asymptotic phase*). The averaged total rewards are further averaged over $500$ Monte Carlo runs for stability. The figure is given in **Figure 7** of the revised manuscript. It is clear that the polynomially decaying step-size is least performing in the initial stage. Moreover, even after $50000$ episodes, its asymptotic phase hasn't kicked in. On the other hand, the fact that $Q$-learning constant learning rate does not converge, is also evident, as larger learning rate constant results in increasing bias. In comparison, both the `LDTZ` and `PDTZ` learning rates maintain a performance comparable to the constant learning rate in the initial phase, while providing convergence in the asymptotic stage.

---

### Official Review · Reviewer_632X · 2025-11-01

**Soundness:** 2
**Presentation:** 2
**Contribution:** 2
**Rating:** 4
**Confidence:** 3

**Summary:**

The paper studies sample complexity of Q-learning under the power-law decay to zero (PD2Z) step-size including LD2Z. The authors prove a convergence rate result using the PD2Z step-size, which has a regime of fast decay and stabilizing phase. The authors also provide a result for tail Polyak Ruppert averages that converges to normal distribution of which the superioity over standard PR is suppported by experimental result. Lastly, an approximation result on the partial sum of the iterates is provided.

**Strengths:**

1. The paper studies the first result of Q-learning using PD2Z step-size. It inherits the fast convergence benefits of a constant step-size with the diminishing error properties of using a polynomial step-size.

2. The paper provides interesting properties not only convergence rate but also tail PR result and strong invariance property.

**Weaknesses:**

1. Importance of studying convergence rate in Q-learning : The literature of Q-learning focused on the sample complexity on $S,A$ or $\frac{1}{1-\gamma}$ dependency, which is lack of here.

2.. The discount factor $\gamma=0.1$ seems to be too small. It differs from usual choice of $\gamma=0.99$.

3.. Figure 4 seems to be an important result for tail PR case but it is deferred to Appendix.

4.. The authors study a result for Q-learning with PD2Z but the unique challenge in applying the particular step-size to Q-learning compared to optimization literature is not clear.

**Questions:**

1. The quantile transformaiton using the uniform random vairable $U$ is not clear. $R$ is defined as a function on $S\times A$ but the authors use $R(s,a,U)$.

2. Can we derive Assumption 3.2 from the definitions?

3. From Figure 1, the performance of constant step-size and LD2Z step-size does not seem to differ much.

---

> ### Author Response · Authors · 2025-11-21
> **Response to Reviewer 632X**
>
> We sincerely thank the reviewer for carefully reading our manuscript and for the positive comments on its interesting results and on acknowledging our novelty. Below, we address your comments individually. We warmly welcome any further suggestions to improve the paper. If our responses adequately address your concerns, we kindly ask you to consider increasing your score.
>
> >**Importance of studying convergence rate in Q-learning:**  The literature on Q-learning has focused on the sample complexity in terms of $|\mathcal{S}|$, $|\mathcal{A}|$, or the dependence on $\frac{1}{1 - \gamma}$, which is lacking here.''
>
> Thank you for raising this important point. We agree that understanding the sample complexity of $Q$-learning with L2ZD step-size schedule is valuable. Motivated by the discussion in Wainwright (2019), we provide an explicit characterization of the *iteration complexity* $N(\epsilon, \gamma, \nu)$ of $Q$-learning with L2ZD schedule. Here, $N(\epsilon, \gamma, \nu)$ denotes the minimal number of samples required to ensure $\|Q_{n, n} - Q^{*}\|_{q} \leq \epsilon$. In this work we focus on the setting with finite state and action spaces, and hence treat $|\mathcal{A}|$ and $|\mathcal{S}|$ as fixed constants. Our primary interest is therefore the dependence of the *iteration complexity* on $\frac{1}{1 - \gamma}$. In particular, taking
>
> $$
> \eta = \frac{1 - \gamma}{(1-\gamma)^2 + 2(p - 1)\gamma^2} \asymp 1 - \gamma
> $$
>
> in Theorem 3.1 yields $c_{3} \asymp (1 - \gamma)$. Consequently, the *iteration complexity* satisfies
>
> $$
> N(\epsilon, \gamma, \epsilon) = O\left(\frac{1}{(1 - \gamma)^{2}} \log \left(\frac{|Q_{0} - Q^{*}|}{\epsilon}\right) + \frac{1}{(1 - \gamma)^{2} \epsilon^{2(\nu + 1)/\nu}}\right).
> $$
>
> We are open to adding this in the main manuscript. We appreciate your comments, which highlight the relevance of extending these results to high-dimensional regimes where $|\mathcal{S}|$ and $|\mathcal{A}|$ may be large. Such an extension is highly nontrivial: high-dimensional strong invariance principles are not well understood even for the partial sum processes of independent sequences, and the classical strong invariance principle may fail when the dimension diverges, particularly since the multivariate CLT itself does not generally hold in high dimensions. For these reasons, we restrict our analysis to finite $|\mathcal{S}|$ and $|\mathcal{A}|$ in the present paper and leave the high-dimensional setting as an important direction for future research.
>
> **Reference:**  Martin J. Wainwright. *Stochastic approximation with cone-contractive operators: Sharp $\ell_{\infty}$-bounds for $Q$-learning.* arXiv preprint arXiv:1905.06265 (2019).
>
> > **The discount factor $\gamma = 0.1$ seems to be too small. It differs from the usual choice of $\gamma = 0.99$.**
>
>    We also ran the experiments for discount factor $\gamma = 0.99$ for $n = 20{,}000$ iterations with step-sizes
>    $\eta_t = 0.25 t^{-0.55}$, $\eta_t = 0.25$, $\eta_t = 0.25(1 - t/n)$, $\eta_t = 0.25(1 - t/n)^2$, and $\eta_t = 0.25(1 - t/n)^3$.
>    The results and interpretations are similar. The figures are provided in Appendix Section 9.
>    We are open to including them in the main manuscript at your suggestion.
>
> > **Figure 4 seems to be an important result for the tail PR case but it is deferred to the Appendix.**
>
>    Thank you for pointing this out. We have moved this to Section 5.4 in the revised manuscript.

---

> > ### Author Response · Authors · 2025-11-21
> > **Ctd.**
> >
> > > **The authors study a result for Q-learning with PD2Z but the unique challenge in applying the particular step-size to Q-learning compared to the optimization literature is not clear.**
> >
> >    We thank the reviewer for raising this pertinent question. There are both crucial similarities and important differences between our results and corresponding results for these step-sizes in the broader optimization literature.
> >
> >    **(i)** To the best of our knowledge, Goldreich et al. (2025) is the first to analyze the LDTZ step-size for SGD. However, their analysis is substantially more limited and, for example, does not include a CLT or a strong invariance principle.
> >
> >    **(ii)** Technically, with moderate amount of effort, one can strive to produce similar results for SGD- which in itself is an interesting direction- but there are some clear differences when it comes to Q-learning. For example, in Q-learning, there is no convex potential whose gradient coincides with Bellman residuals, which prevents construction of appropriate Lyapunov functionals.
> >
> >   Our original motivation for this paper is its wide application in LLM training via RLHF optimization (eg. Devlin et. al. 2019). Therefore, naturally, we chose to work with Q learning as the simplest such RL algorithm for which the theoretical results were unknown. Similar techniques, albeit with non-trivial, case-by-case modifications can conceivably be applied to derive corresponding results for general stochastic optimization algorithms, but they nevertheless carry the same take-home message about the usefulness of these learning rate choices, that we provide in this paper. Nevertheless, these constitute areas for interesting theoretical development, and we thank the reviewer for pointing us in these directions.
> >
> >    **Reference:**
> >    Jacob Devlin, Ming-Wei Chang, Kenton Lee, and Kristina Toutanova. *BERT: Pre-training of deep bidirectional transformers for language understanding.* In NAACL-HLT 2019.
> >
> >
> > > **The quantile transformation using the uniform random variable $U$ is not clear. $R$ is defined as a function on $\mathcal{S} \times \mathcal{A}$ but the authors use $R(s,a,U)$.**
> >
> >    We sincerely apologize for the confusion. Our intuition is as follows: since $R(s,a)$ is a random variable, there exists a function $F_{s,a}(\cdot)$ (for example, the inverse CDF of $R(s,a)$) and a uniform random variable $U \sim \mathrm{Unif}[0,1]$ such that
> >    $
> >    R(s,a) = F_{s,a}(U) \quad \text{almost surely}.
> >    $
> >    This follows from the following simple result: for any random variable $X$ with CDF $F_X$, there exists a uniform random variable $U \sim \mathrm{Unif}[0,1]$ such that
> >    $
> >    F_X^{-1}(U) = X.
> >    $
> >    With a slight abuse of notation, we therefore write
> >    $
> >    R(s,a,U) := F_{s,a}^{-1}(U).
> >    $
> >
> > >  **Can we derive Assumption 3.2 from the definitions?**
> >
> >    Thank you for raising this great question. After a slight thought, we realized that the key insight behind the Assumption 3.2 is ensuring that the optimal policy, and the optimal quality function is unique. In that regard, even though Assumption 3.2 cannot be directly derived from the conditions stated, it can indeed be obtained via a simple margin condition, that can be more illuminating in the Q-learning context.
> >
> >    Specifically, let the greedy policy $\pi^\star(s) = \arg\max_a Q^\star(s,a)$ be unique for every state $s$, such that
> >    $$ \Delta = \min_s (Q^\star(s,\pi^\star(s)) - \max_{a \ne \pi^\star(s)} Q^\star(s,a )) > 0. $$
> >
> >  Under this condition, we derive Assumption 3.2. To that end, suppose
> > $|\mathbf{Q} - \mathbf{Q}^\star|\_{\infty} \leq \Delta/2.$
> > Then, by definition of the greedy policy,
> > $
> > H^{\pi\_Q} = H\_{\pi^\star},
> > $
> > and hence, Assumption 3.2 is trivially satisfied. On the other hand, if $|\mathbf{Q} - \mathbf{Q}^\star|\_{\infty} > \Delta/2$, then from $|H^{\pi_Q} - H^{\pi^\star}|\leq 2$ it follows
> > $$
> > |(H^{\pi_Q}-H^{\pi^\star})(\mathbf{Q}-\mathbf{Q}^\star)|\_{\infty} \leq 2 |\mathbf{Q}-\mathbf{Q}^\star|\_{\infty} \leq \frac{4}{\Delta} |\mathbf{Q}-\mathbf{Q}^\star|\_{\infty}^2.
> > $$
> > Thus we can derive Assumption 3.2 from a basic margin condition. We are open to adding this into the main manuscript at your suggestion.
> >
> > > **From Figure 1, the performance of constant step-size and LD2Z step-size does not seem to differ much.**
> >
> > Although Figure 1 shows similar early performance for the constant and LD2Z step-sizes, their long-run behavior differs crucially. With a constant step-size, the Q-iterates do not converge to $\mathbf{Q}^\star$; instead, they converge to a stationary distribution and retain a persistent bias, as seen in the noisy green curves on the right panel. In contrast, the LD2Z schedule preserves the fast initial decay of constant step-sizes while ensuring convergence to $\mathbf{Q}^\star$, as reflected in the orange curves in the right panel of Figure 1 and in other figures throughout the paper. This can also be seen in Figure 7 of the revised manuscript.

---

> ### Comment · Reviewer_632X · 2025-11-27
>
> Thank you for the detailed response.
>
> - The known minimax sample complexity of tabular Q-learning scales at the order of $1/(1-\gamma)^4$ whereas the one authors derived is $1/(1-\gamma)^2$. Where does this gap come from?
> - Even though |S||A| may be constants (where I initially mentioned in the context of uniform sampling of each state-action pairs), the dependence on the minimum state–action visitation probability still plays an important role and cannot be ignored.
>
>
> Li G, Cai C, Chen Y, et al. Is Q-learning minimax optimal? a tight sample complexity analysis[J]. Operations Research, 2024, 72(1): 222-236.

---

> ### Comment · Reviewer_HuHH · 2025-11-27
>
> The discrepancy might arise from comparing bounds in different settings (minimax high-probability $\ell_\infty$ sample complexity vs. non-asymptotic $\ell_p$ error bounds with problem-dependent constants, where part of the $(1-\gamma)$-dependence is hidden). This is an important point, I suggest the authors restate their main results in a more “sample-complexity-style” form and make the $(1-\gamma)$-dependence of the constants explicit so the comparison to the minimax $1/(1-\gamma)^4$ rate is transparent.

---

> > ### Author Response · Authors · 2025-11-29
> > **Reply to reviewers 632X and HuHH**
> >
> > Thank you for the detailed response and the opportunity to clarify further.
> >
> > > The known minimax sample complexity of tabular $Q$-learning scales at the order of $1/(1-\gamma)^{3}$ whereas the one authors derived is $1/(1-\gamma)^{2}$. Where does this gap come from?
> >
> >   **Response:** Thank you for pointing out this gap. We apologize for the oversight in our previous response, and we also appreciate Reviewer HuHH for the insightful comment highlighting the need of further analysis on problem-dependent constants. Our error stemmed from not accounting for the dependence of the $p$-th moment $\Theta_{p}$ in Theorem 3.1. By incorporating this dependence ($\Theta_{p}^{1/p} \lesssim \frac{1}{1 - \gamma}$), we obtain the following *iteration complexity*:
> >
> >   $$
> >   N(\epsilon, \gamma, \epsilon)
> >   = O\left(
> >       \frac{1}{(1 - \gamma)^{2}}
> >       \log \left(\frac{|Q_{0} - Q^{*}|}{\epsilon}\right)
> >       +
> >       \frac{1}{(1 - \gamma)^{4 + 2/\nu}
> >       \epsilon^{2(\nu + 1)/\nu}}
> >     \right).
> >   $$
> >
> >   We note that for large $\nu$, the rate approximately matches that derived by [1]. The gap for a finite value of $\nu$ can also be explained by the much weaker assumption that we work with. For example, we do not assume the rewards to be bounded, and therefore, are only constrained to work with finite $p$-th moments of the Bellman noise. In contrast, [1] assumes the rewards $\in [0,1]$, which makes the Bellman noise sequences bounded and allows them to use finer tools from subGaussian theory, such as Freedman's inequality (in contrast to the Burkholder's inequality which is sharp in absence of boundedness). It is conceivable that in presence of stricter assumption, the worst-case sample complexity can be further improved, but that is non-trivial. It is also evident from Theorem 3.3 that the sample complexity likewise derived based on Polynomially decaying rate will be worse compared to the PD2Z rates, unless we take the minimax optimal choice $\eta_t\approx 1/t$. We believe that further analysis to achieve *sharper results with stronger assumptions* constitutes an interesting direction of future work, especially in light of the key message in this work being the efficacy of LD2Z and PD2Z rates over its polynomial and constant counterparts.
> >
> >   **Reference:**
> >   [1] Li G, Cai C, Chen Y, et al. *Is Q-learning minimax optimal? A tight sample complexity analysis.* Operations Research, 2024.
> >
> >
> >
> >
> > > Even though $|\mathcal{S}| |\mathcal{A}|$ may be constants (where I initially mentioned in the context of uniform sampling of each state-action pairs), the dependence on the minimum state-action visitation probability still plays an important role and cannot be ignored.
> >
> >   **Response:** We again thank the reviewer this insightful comment. Indeed, state-action interaction is important in Q-learning, especially with regards to sample complexity and convergence. We note, however, that our initial focus is on statistically valid inference, and we respectfully point out that in this literature, the known statistical inference results such as central limit theory for Q-learning and temporal difference learning [1,2,3,4,5] usually assume finite $|\mathcal{S}||\mathcal{A}|$ and do not track the relevant constants. This is mostly due to the fact that in the cases where tracking $|\mathcal{S}||\mathcal{A}|$ is important, i.e. when they are not constants, it requires applications of significantly different tools. Nevertheless, it is indeed possible to understand the role of $|\mathcal{S}||\mathcal{A}|$ through the light of sample complexity and $\mathcal{L}_p$ convergence. For example, noting that
> >
> >   $$
> >   \Theta_p^{1/p} \lesssim \frac{|\mathcal{S}||\mathcal{A}|}{1 - \gamma},
> >   $$
> >
> >   one can incorporate this quantity in the sample complexity. In presence of boundedness, this can potentially be improved further.
> >
> >   **References:**
> >   [1] Li, Xiang, et al. *A statistical analysis of Polyak-Ruppert averaged Q-learning.* AISTATS, 2023.
> >   [2] Panda, Saunak Kumar, et al. *Online Statistical Inference of Constant Sample-averaged Q-Learning.* RL Safety Workshop.
> >   [3] Mou, Wenlong, et al. *On linear stochastic approximation: Fine-grained Polyak-Ruppert and non-asymptotic concentration.* COLT, 2020.
> >   [4] Samsonov, Sergey, et al. *Gaussian approximation and multiplier bootstrap for Polyak-Ruppert averaged linear stochastic approximation with applications to TD learning.* NeurIPS, 2024.
> >   [5] Samsonov, Sergey, et al. *Statistical inference for Linear Stochastic Approximation with Markovian Noise.* arXiv:2505.19102 (2025).

---

### Author Response · Authors · 2025-11-27
**General Comments**

We thank all reviewers for their careful reading of our submission and for their thoughtful and constructive feedbacks. In particular, we appreciate that the reviewers acknowledged the following strengths of our work:

- Reviewers **632X**,  and **HuHH** both noted that the paper delivers the first non-asymptotic (or first rigorous) analysis of Q-learning under PD2Z / LD2Z–type step-size schedules, which had not been previously developed. Additionally, reviewers **WVUn** and **vcVR** appreciated the comprehensiveness of  ourtheoretical guarantees, including convergence rates, asymptotic normality of Polyak–Ruppert averaging, and strong approximation results.

- Reviewers **632X, HuHH, and WVUn** concurred with our point that the results capture both fast initial behavior (as with constant step sizes) and favorable long-run properties (as with polynomial decay), and that the analysis formally explains these two regimes.

- Reviewers **HuHH, WVUn, and vcVR** emphasized that the paper is clearly written and logically structured, with explicit assumptions and presentation that aligns with classical stochastic-approximation frameworks. The reviewers are also in consensus that the empirical results support the theoretical claims, illustrating the relative advantages of different step-size strategies and corroborating the predicted behavior.

- Reviewer **WVUn** specifically noted that LD2Z/PD2Z has practical relevance for real-world applications (e.g., generative models) and that the paper’s theoretical development helps clarify why such schedules are attractive.

Moreover, we have carefully addressed all comments raised by the reviewers. Accordingly, **we have also revised the manuscript with a few additional experiments and discussions based on the reviews**. In the revised manuscript, all corresponding updates have been incorporated and are highlighted in red for ease of reference. We hope that our work will be evaluated favorably in light of its originality, technical novelty, potential impact, and the revisions made in response to the review. The detailed feedback not only helped us address potential ambiguities but also inspired meaningful improvements to the quality of the final paper. As such, we are happy to engage with any further comments and questions that the reviewers might have to further improve the paper.

# **Update**

In the context of the present circumstances, we also provide a concise summary of the key changes and improvements made during the rebuttals for the convenience of the AC.

- **Reviewer 632X** asked about our sample complexity results and the dependence on $\mathcal{S}$, $\mathcal{A}$, and $\frac{1}{1-\gamma}$. We addressed these in the rebuttal and in **Remark 3.1**, showing that our iteration complexity approximately matches the known minimax rate for tabular Q-learning. We also clarified the dependence on $\mathcal{S}$ and $\mathcal{A}$.

- **Reviewer 632X** also questioned the necessity of Assumption 3.2. We added a discussion in **Appendix Section 9** showing how a simple margin condition implies this assumption.

- As suggested by **Reviewer 632X**, we added experiments with $\gamma = 0.99$ in **Appendix Section 10**.

- Following **Reviewer HuHH**’s suggestion, we added **Figure 7** (Section 9.1), reproducing Figure 6.9 from Sutton & Barto (2018) for our experimental setup. This further validates the superiority of LDTZ and PDTZ rates over polynomial and constant learning rates.

- **Reviewer WVUn** inquired about asymptotic variances under LD2Z and polynomial step-sizes. We ran additional simulations to address this and included the results in the rebuttal.

- **Reviewer vcVR** requested more experimental details and additional linearly decaying step-size experiments. These are now included in **Figures 5–6, Section 10**, strengthening our empirical conclusions.

- **Reviewer vcVR** also raised a concern about L2ZD making an online algorithm effectively offline. We clarified that the method remains online and only requires an estimate of the data size (and not an offline repository of samples themselves), which often depends on some notion of cost you are willing to spend. Building on, we proposed an alternative (but similar to LDTZ) step-size that avoids using $n$ (with theoretical analysis left for future work).

- Some general clarification questions were on how the results are different compared to previous stochastic approximation/SGD literature (Reviewers **632X** and **WVUn**), and why the covariance matching approximation (Thm 4.2) is better than the Brownian motion approximation (Reviewer **vcVR**). These have been clarified at the appropriate places.

Finally, we are pleased to note that the reviewers were unanimous in their satisfaction with our detailed response and acknowledged that we addressed most of their concerns. We are thankful for their active engagement and constructive suggestions on our work.

---

### Meta-Review · Area_Chair_W445 · 2026-01-09

**Summary:**

This paper presents a sharp asymptotic theory for Q-learning using linear decay to zero (LD2Z) and power-law decay to zero (PD2Z-$\nu$) learning rate schedules. The authors establish that these schedules achieve a "best-of-both-worlds" property, inheriting the rapid initial convergence of constant step-sizes while maintaining the asymptotic stability of polynomially decaying rates. Key technical contributions include sharp non-asymptotic error bounds for the iterates, the development of a novel tail Polyak-Ruppert averaging estimator with proven asymptotic normality, and a strong invariance principle that facilitates bootstrap-based inference through time-uniform Gaussian approximations. Extensive numerical experiments on a gridworld environment validate these results, demonstrating that LD2Z significantly outperforms traditional polynomial schedules in reaching the vicinity of the optimal Q-function.

**Reviewer Concerns:**

The rebuttal effectively addressed several major technical criticisms regarding minimax optimality, empirical validation, and theoretical assumptions. Reviewer HuHH and Reviewer 632X both identified a significant gap between the authors' initial iteration complexity of $1/(1-\gamma)^2$ and the known minimax rate of $1/(1-\gamma)^4$. The authors successfully resolved this by acknowledging an oversight and providing a corrected complexity bound that aligns with known minimax results. Furthermore, the authors responded to concerns about the limited scope of their initial experiments by providing new simulations using a standard discount factor of $\gamma=0.99$ and reproducing classic gridworld benchmarks from the Sutton & Barto textbook to prove the superiority of the LD2Z schedule. They also satisfied Reviewer 632X's query regarding Assumption 3.2 by deriving it from a more intuitive margin condition in the updated appendix. The reviewers also raised significant questions about the online applicability and asymptotic variance of the proposed rates. Reviewer vcVR argued that the dependency of the LD2Z schedule on the total sample size $n$ makes it unsuitable for online learning. The authors mitigated this concern by demonstrating that their asymptotic results remain valid even if $n$ is misspecified and by proposing a potential piecewise schedule for future online streaming use. Regarding asymptotic variance, the authors admitted to Reviewer WVUn that while polynomial decay is theoretically more stable in the long run, LD2Z is practically superior because the asymptotic regime for polynomial rates often requires an unrealistically large number of iterations to kick in.

However, a few concerns remain outstanding for future research. The explicit dependence of the sample complexity on the state and action space sizes ($S$ and $A$) was not fully incorporated into the main results; the authors noted that extending strong invariance principles to high-dimensional regimes is "highly nontrivial" and beyond the current scope. Additionally, the mathematical trade-off between last-iterate convergence and asymptotic normality remains a point of divergence. The authors acknowledged that their scheduler cannot achieve the $\sqrt{n}$ scaling efficiency of standard Polyak-Ruppert averaging for finite $\nu$, though they argued that the massive speed-up in the initial phase justifies this sub-optimality. Finally, while a piecewise version of LD2Z was suggested to handle unknown data sizes, the rigorous theoretical analysis for such an online method was explicitly left for future work.

**Reviewer Scores:**

- **Reviewer 632X:** This reviewer was initially concerned about the missing sample complexity analysis and the lack of clarity regarding the margin condition for Assumption 3.2. Since the authors provided a corrected iteration complexity formula that matched minimax rates and added a detailed derivation of the assumption in the appendix, the reviewer would likely have moved to a 6.

- **Reviewer HuHH:** This reviewer actively pushed for better empirical evidence and a more transparent comparison of the $(1-\gamma)$ dependence. The authors' successful reproduction of classic gridworld benchmarks from the Sutton & Barto textbook and the refinement of the sample complexity to match the $1/(1-\gamma)^4$ rate directly addressed these requests. This reviewer would likely have maintained their score or upgraded it to an 8.

- **Reviewer vcVR:** This reviewer raised the most critical points regarding the "offline" nature of the schedule and the trade-off between last-iterate convergence and normality. The authors' demonstration of robustness to $n$ misspecification and the introduction of a bootstrap algorithm for inference led the reviewer to state that most of their concerns have been resolved, suggesting that the reviewer would have raised their score to a 6.

---

### Decision · Program_Chairs · 2026-01-26

Accept (Poster)